ecology, environmental science

ecosystem metabolism, organic matter, hippopotamus, livestock, subsidy, primary production

**Author for correspondence:**
Frank O. Masese
e-mail: fmasese@uoeld.ac.ke,
f.masese@gmail.com

# Hippopotamus are distinct from domestic livestock in their resource subsidies to and effects on aquatic ecosystems

Frank O. Masese[1,2], Mary J. Kiplagat[1], Clara Romero González-Quijano[2], Amanda L. Subalusky[3,4], Christopher L. Dutton[4], David M. Post[4] and Gabriel A. Singer[2,5]

[1]Department of Fisheries and Aquatic Sciences, University of Eldoret, P.O. Box 1125-30100, Eldoret, Kenya
[2]Department of Ecohydrology, Leibniz-Institute of Freshwater Ecology and Inland Fisheries (IGB), Berlin, Germany
[3]Department of Biology, University of Florida, Gainesville, FL 32611, USA
[4]Department of Ecology and Evolutionary Biology, Yale University, 165 Prospect Street, New Haven, CT 06511, USA
[5]Department of Ecology, University of Innsbruck, Technikerstrasse 25, A-6020 Innsbruck, Austria

FOM, 0000-0002-5912-5049; CRG-Q, 0000-0002-7343-5561; ALS, 0000-0001-8893-6871; CLD, 0000-0002-1091-7571; DMP, 0000-0003-1434-7729; GAS, 0000-0002-7389-9788

In many regions of the world, populations of large wildlife have been displaced by livestock, and this may change the functioning of aquatic ecosystems owing to significant differences in the quantity and quality of their dung. We developed a model for estimating loading rates of organic matter (dung) by cattle for comparison with estimated rates for hippopotamus in the Mara River, Kenya. We then conducted a replicated mesocosm experiment to measure ecosystem effects of nutrient and carbon inputs associated with dung from livestock (cattle) versus large wildlife (hippopotamus). Our loading model shows that per capita dung input by cattle is lower than for hippos, but total dung inputs by cattle constitute a significant portion of loading from large herbivores owing to the large numbers of cattle on the landscape. Cattle dung transfers higher amounts of limiting nutrients, major ions and dissolved organic carbon to aquatic ecosystems relative to hippo dung, and gross primary production and microbial biomass were higher in cattle dung treatments than in hippo dung treatments. Our results demonstrate that different forms of animal dung may influence aquatic ecosystems in fundamentally different ways when introduced into aquatic ecosystems as a terrestrially derived resource subsidy.

## 1. Introduction

The transfer of organic matter from terrestrial to aquatic environments has often been understood to be dominated by litterfall and hydrologic transfers during storms and precipitation events [1,2]. However, it is increasingly recognized that large mammalian herbivores (LMH) can be major agents of transfer of terrestrial organic matter and nutrients into aquatic ecosystems [3,4]. While rates vary widely over broad spatial and temporal scales and depend on the characteristics of the animal vector and the recipient ecosystem [5,6], the amount can be significant, especially for low-order streams in rangelands and pastoralist areas [3,7,8].

Terrestrial and aquatic ecosystems in many African savannah landscapes are intricately linked via the vectoring role that LMH play in transferring large amounts of resources from terrestrial to aquatic ecosystems [9,10]. Pathways of organic matter and nutrient input into aquatic ecosystems by LMH include egestion and excretion during migrations and watering [11], facilitation of soil erosion [9] and drowning during water crossings [4]. A prominent example is the common hippopotamus (*Hippopotamus amphibius*, hereafter hippo), which migrates daily between savannah grasslands, where it forages, and aquatic ecosystems where it rests and much of its defaecation occurs

[12]. Resource subsidies from hippos alter primary production and secondary production, most prominently through direct consumption by bacteria, invertebrates and fish [13–15], and influence whole river oxygen dynamics, biogeochemistry and community composition [7,8,16].

The expansion of human settlements, crop farming and conversion of forests and savannah grasslands to pasture for livestock production have contributed to the loss of large populations of wild LMH around the world [17–19]. In many African savannahs, large populations of wild herbivores still dominate the biomass of conservation areas [20,21]. However, even in these regions, wild LMH are declining concurrently with increases in livestock such as cattle, goats and sheep [18,22]. In most areas where livestock have replaced wildlife on the landscape, their influence on aquatic systems has often been seen as negative, with research focusing on habitat degradation, nutrient and organic matter loading and microbial contamination [3,23]. However, livestock may take over some of the ecological roles historically filled by wild LMH, thereby maintaining the functionally important linkage of riverine ecosystems to their surrounding terrestrial landscapes. The degree to which ecosystem effects of this functional linkage from livestock are similar to those from wild herbivores depends in part on the similarity of the resource subsidies they transport.

Ruminants such as cattle and sheep have a relatively efficient digestive system compared with non-ruminants such as hippos and horses, and this difference in digestion produces smaller faecal particle sizes in ruminants [24,25]. Non-ruminants, such as hippos, have longer mean retention times than ruminants, which enhances nutrient extraction from ingesta, leading to a greater ratio of C to nutrients, reflecting relatively lower quality of their dung (electronic supplementary material, table S1). Ruminants also forage on a broader selection of plant species compared with non-ruminants [26,27], and by so doing they ingest a wider variety of metabolites and chemicals [28], which may result in differences in the chemical composition of dung and its leachate, and consequently its effect on ecosystem processes.

Differences in particle size and composition are likely to influence the way in which dung inputs from ruminants and non-ruminants influence aquatic ecosystems. Dung comprising large particles with a high ratio of C to nutrients, as expected from non-ruminants such as hippos (electronic supplementary material, table S2), is qualitatively similar to the seasonal input of leaf litter to aquatic ecosystems in temperate forests [15,29]. These inputs are expected to deposit in the benthos as relatively refractory material that increases ecosystem respiration and is incorporated into the detrital food web. Dung comprising small particles that are relatively high in nutrients, as expected from ruminants such as cattle, is expected to remain suspended in the water column, which could decrease light penetration, but also be more likely to increase both water column and benthic primary production [30]. Furthermore, the addition of nutrient-rich ruminant dung from cattle to aquatic ecosystems already receiving large inputs of carbon-rich non-ruminant dung from hippos may lead to interactions between the two subsidies in decomposition rates and ecosystem effects [31]. The incremental displacement of wild herbivore populations by livestock raises the question of how this change may impact the transfer of nutrients and organic matter into inland waters and the ensuing ecosystem responses.

The Mara River and its seasonal tributaries in the Maasai Mara National Reserve (MMNR) in Kenya host more than 4000 hippos [32]. There are also over 250 000 cattle in communal lands adjoining the MMNR, where livestock coexist with wildlife [33]. This distribution results in a displacement pattern with hippo areas inside the reserve, mixed hippo and livestock areas outside the reserve and only livestock grazing areas further away from the reserve [34]. This overlapping distribution of livestock and wildlife raises the question of how aquatic ecosystems will respond to the displacement of wild LMH by livestock as agents of resource transfer from terrestrial to aquatic environments.

Here, we characterize the particle size and stoichiometry of cattle and hippo dung, estimate the loading of organic matter by cattle and hippos into the Mara River and conduct an experiment in recirculating experimental stream mesocosms to test the impacts of these different inputs on the function of aquatic ecosystems. Previous research has already shown that the quantity of inputs by LMH has substantial impacts on the aquatic ecosystem [15,16]. For our experiment, we used a replacement design to compare ecosystem effects of cattle dung and hippo dung both independently and in combination with one another. We measured the effects of both hippo and cattle dung on nutrients, dissolved organic carbon (DOC) quantity and quality, gross primary production (GPP) and ecosystem respiration (ER). We hypothesized that cattle dung inputs would lead to higher nutrient concentrations and increased GPP, while hippo dung inputs would lead to higher concentration of DOC and increased ER. Furthermore, we hypothesized that these parameters may change nonlinearly along a gradient of low to high subsidy quality—that is, from a system dominated by hippo dung to one dominated by cattle dung—depending on the strength of the interaction between the high C and high nutrient inputs. We further hypothesized that cattle dung would lead to a more diverse DOC composition given the broad foraging strategy of cattle compared with hippos.

## 2. Material and methods

### (a) Characteristics of cattle and hippo dung

Macro- and micronutrient composition of cattle and hippopotamus faecal samples were analysed at the Leibniz Institute for Zoo and Wildlife Research, Berlin, Germany. Before analysis, dried samples were ground to a particle size of about 1 mm. For C and N, samples were weighed and analysed on an elemental analyser (Hekatech, Thermo Finnigan). For P, samples were weighed, ashed in a muffle furnace at 550°C and then digested before analysis on an inductively coupled plasma optical emission spectrometer (ICP-OES) (PerkinElmer, Ueberlingen, Germany). Crude protein was calculated as 6.25 × N [35]. Carbohydrates (sucrose, D-glucose, D-fructose and starch) were analysed using commercial enzymatic test kits from R-Biopharm (Darmstadt, Germany). For mineral analysis (Ca, Mg, Fe, K), samples were microwave digested and analysed by a PerkinElmer ICP-OES.

### (b) Livestock versus hippopotamus loading rates of organic matter

We developed a model to estimate cattle loading rates of organic matter (dung) into the Mara River (electronic supplementary material §S1) and compared results with existing estimates of

loading rates for hippos in the river extracted from Subalusky *et al.* [12]. We used literature to estimate daily dry matter intake (DMI) of zebu cattle, and the proportion of organic matter (OM) egested or excreted. We estimated cattle loading rates of OM as a fraction of time spent in the river, and we multiplied the per capita loading rate by the cattle population to get the total loading rates for all cattle. We then compared the loading of cattle and hippopotamus dung in two areas of the Mara River where their distributions overlap.

## (c) Experimental mesocosms

We used experimental stream mesocosms constructed out of PVC canvas measuring 4.2 m long and 19 cm wide [15]. Water was recirculated in each mesocosm by paddlewheels affixed to a shaft that was powered by a motor, with each of three shafts handling six streams (electronic supplementary material §S2). We had three replicates for each of six dung treatments in a replacement design ranging from 100% hippo dung to 100% cattle dung with 20% increments of replacement (electronic supplementary material, figure S1). This approach allowed us to test for potential interactive effects between dung types, recognizable by nonlinear responses to the dung treatment gradient. Treatments were randomly distributed among mesocosms, with a replicate of each treatment in each of the three blocks. A total of 120 g (wet weight, 1.7 g l$^{-1}$) of dung was distributed in each mesocosm once at the beginning of the experiment in order to study ecosystem responses arising from differences in dung quality due to nutrient leaching and mineralization rates.

To accelerate biofilm growth, mesocosms were inoculated with periphyton scraped off rocks from the Amala River, a tributary of the Mara upstream of wildlife. Each mesocosm was lined with six unglazed ceramic tiles that were used for weekly sampling of biofilms. Each week, one tile from each mesocosm was destructively sampled without replacement for analysis of ash-free dry mass (AFDM).

## (d) Water sampling and analysis

We collected water samples weekly, including day 1, for analysis of ammonium ($NH_4^+$), soluble reactive phosphorus (SRP), total phosphorus (TP), nitrite ($NO_2^-$), nitrate ($NO_3^-$), total suspended solids (TSS), particulate organic matter (POM), dissolved organic carbon (DOC) concentration and composition, and chlorophyll *a* (Chl-*a*). Further details on analysis of nutrients and DOC, Chl-*a*, TSS and POM concentrations are available in electronic supplementary material §S3.

We characterized DOC by absorbance and fluorescence analyses, which provide proxies for DOC source and/or biological availability [36,37]. To characterize DOC, we used parallel factor analysis (PARAFAC) to decompose 349 excitation–emission matrices (EEMs) into fluorescent components [38], and size-exclusion chromatography (SEC) [39], which separates three size fractions: humic substances (HS), high molecular weight non-humic substances (HMWS) and low molecular weight substances (LMWS). Further details on collection and analysis of DOC composition data are available in the electronic supplementary material §S4.

## (e) Metabolism

In each mesocosm, we recorded dissolved oxygen (DO) and water temperature every 1 min for six weeks with MiniDOT loggers (PME, Vista, CA, USA). Light intensity (0 to 320 000 lux) was recorded with HOBO Pendant Temperature/Light Data Loggers (UA-002-64; Onset, Bourne, MA, USA). We then estimated GPP and ER from diel changes of DO, temperature and irradiance using an inverse modelling procedure that included temperature-dependent ER, light-dependent GPP and reaeration [40]; details are provided in electronic supplementary material §S5.

## (f) Data analysis

We used linear mixed effect models (LMEMs) to test the effect of dung treatment on response variables DOC, Chl-*a*, AFDM, TSS, POM, SRP, $NH_4^+$, $NO_2^-$ and $NO_3^-$ with the lme function in the 'nlme' package [41] in R [42]. LMEMs were used after residuals displayed linear responses to dung treatment. LMEMs included dung treatment (six levels) and time (week 0 to 6) as fixed effects and block as a random effect. We also included an interaction of dung treatment with time to test for differences in the temporal dynamics of responses. Response variables were log-transformed when appropriate to meet normality assumptions. We ran a separate model for each variable and computed marginal $R^2$ ($R_m^2$, variance explained by fixed factors) and conditional $R^2$ ($R_c^2$, variance explained by the entire model, i.e. by fixed and random factors) coefficients with the r.squaredGLMM function in the MUMIN package [42].

To test the effect of cattle dung on ecosystem respiration, or production, we fitted a three-parameter sigmoid Gompertz model [43], given by

$$Y(t) = K \cdot e^{-\text{lag} \cdot e^{-\text{rate} \cdot t}},$$

to daily GPP and ER for each dung treatment (i.e. with data pooled across three streams) and separately for each replicate stream; this yielded estimates for the upper asymptote ($K$), growth rate (rate) and a dimensionless parameter for location along the time axis (lag), which shifts the graph to the left or right and is related to the time taken to reach the upper asymptote (maximum GPP or ER), with high values indicating faster progression towards maximum GPP or ER. $Y(t)$ is the expected value (GPP or ER) as a function of time (days since start of the experiment) and $t$ is time in days. These parameter estimates were then regressed against dung treatment (% cattle dung) using general additive mixed modelling (GAMM) to avoid strong assumptions about potential relationships. GAMMs were built using penalized cubic regression splines with degrees of freedom automatically identified based on the generalized cross-validation score (GCV). Further, to investigate weekly changes in ecosystem metabolism (GPP, ER, GPP/ER and net ecosystem production (NEP)), weekly means for each stream were computed (total six weeks). We then tested for differences among dung treatments using GAMMs [44], and included dung treatment as a fixed effect and block as a random effect. GAMMs were fitted using the R package mgcv [45].

Principal component analysis (PCA) was used for dimension reduction of DOC quality data (absorbance- and fluorescence-derived indices FIX, $\beta/\alpha$, humification index (HIX); PARAFAC components C1 to C7; SEC results HMWS (in %), HS (in %), LMWS (in %)). While optical indices and SEC results are expressed as ratios or percentages and thus describe composition with little or no influence of DOC quantity, PARAFAC components were used in the form of quantitatively reliable absolute fluorescence intensities. All variables were scaled to zero mean and unit s.d. prior to use in PCA. Statistical analyses were performed with R version 3.3.1 [42] using the packages vegan [46], sem [47] and deSolve [48].

# 3. Results

## (a) Characteristics of cattle and hippo dung

Cattle dung had lower C : N : P ratio than hippo dung (electronic supplementary material, table S3). C : N : P was 155.2 : 5.1 : 1.0 for cattle dung and 261.4 : 7.6 : 1.0 for hippo dung (electronic supplementary material, table S3); per unit C, cattle dung was thus richer in N and P than hippo dung by 13 and 69%, respectively. Further, cattle dung was

enriched in the micronutrients Ca, Fe, K and Mg by 3.6–31% (electronic supplementary material, table S3).

## (b) Loading rates of organic matter by cattle and hippopotamus

We estimated that cattle spend 10 min in the Mara River or tributaries per day, and each on average loads 22.3 g DM (86.6 g wet mass) (kg body mass)$^{-1}$day$^{-1}$ into the river (electronic supplementary material §S1). Thus, an average animal (265 kg) defaecates 12.5 kg faeces (wet mass) every day, and 0.0866 kg (0.70% of daily defaecation) goes into the Mara River. In comparison, an average hippo (1500 kg) defaecates 17.4 kg faeces (wet mass) every day, and 8.7 kg (50%) goes into the Mara River [12].

Using cattle population estimates [22,33,49], we estimated the total daily loading was 1157 kg faeces into the Mara River inside the MMNR, 2599 kg outside the MMNR and 7364 kg along the Talek River. Within the MMNR, livestock loading is only around 6% of loading due to cattle and hippopotamus because cattle are not supposed to be grazed in the reserve, but illegal grazing of a small number of cattle nevertheless occurs. Outside the MMNR, where the Maasai pastoralists keep large numbers of cattle, loading by cattle along the Mara and Talek rivers increases to nearly 16 and 57%, respectively, of the total organic matter loading due to cattle and hippopotamus. These loading rates are based on the assumption that all cattle within either the Mara or Talek sub-catchment visited the river for watering or crossing at least once per day. However, some cattle may use water pans for their water needs during certain portions of the year.

## (c) Nutrients

Dung treatment had a significant effect on nutrient concentrations, with a significant increase in SRP, ammonium, nitrite and nitrate with increasing proportion of cattle dung (table 1; electronic supplementary material, figure S2). There was also a significant effect of time on nutrient concentrations, reflecting different rates of leaching, uptake and retention in biomass (table 1). Notably, there was a greater than 90 and 56% reduction in SRP and ammonia, respectively, within the first two weeks across all treatments (electronic supplementary material, figure S3a,b). Similarly, nitrite significantly declined after the second week, while nitrate increased (electronic supplementary material, figure S3c,d; table 1).

## (d) Organic matter and biomass

Dung treatment had a significant effect on organic matter (DOC and POM), TSS, water column Chl-a and biomass of biofilms (AFDM) (table 1). There was a linear increase in these variables from low to high proportion of cattle dung (electronic supplementary material, figure S4; table 1). DOC concentration was considerably higher with the presence of cattle dung through the entire experiment (electronic supplementary material, figure S4a). There was also a significant effect of time on these variables (table 1). DOC concentrations increased by greater than 50% over the experimental period (electronic supplementary material, figure S4a), and Chl-a, AFDM, TSS and POM increased by greater than 100% (electronic supplementary material,

figure S4b–e). Further, we observed that the smaller particles of cattle dung [24,25] remained suspended in water while those from hippo dung sank to the bottom, which was reflected by higher TSS and POM in the water column with higher proportions of cattle dung (electronic supplementary material, figure S4d,e).

## (e) DOC composition

The PARAFAC model consisted of seven components (electronic supplementary material §S4, table S4 and figures S5 and S6): four humic-like, one reduced humic-like and two protein-like fluorescence components. The first two PCA axes explained 32.5 and 27.0% of the total variance, respectively, and efficiently depicted treatment differences and development of DOC composition throughout the experimental time (figure 3). The 100% cattle treatment was clearly separated from all other treatments and furthest from the 100% hippo treatment, in particular along PC1. By contrast, PC2 was more important for capturing temporal changes, but also contributed to definition of distinct treatment-specific DOC composition at the experiment start. At the start of the experiment, all dung treatments produced DOC with the highest share of low molecular weight substances and rich in aromatic structures and humic substances indicative of leaching from plant material (figure 3a,b). DOC from cattle dung was more fluorescent and humic compared with hippo dung, which, by contrast, had seemingly fresher DOC with relatively N-deficient high molecular weight substances (figure 3a,b). Over the duration of the experiment DOC composition in all dung treatments changed in parallel towards a common endpoint of higher concentrations of less fluorescent, less humic and less aromatic DOC (figure 3a,b). High molecular weight substances with high C : N, likely carbohydrates from primary production, became more important towards the end of the experiment. Notably, DOC in the 100% cattle dung treatment experienced a strong and long-lasting phase of humic fluorescence buildup before rejoining the other treatments' trend. In an effort to quantify overall compositional dynamics, we summed Euclidean path lengths from the start to the end of the experiment in the multivariate space described by all PCs. While the 0 and 20% cattle dung treatments here resulted in DOC with minimal turnover, the 60% cattle treatment had the highest compositional turnover of DOC (figure 3c). This suggests the sum of two processes—leaching from the dung and autochthonous production—cause very dynamic DOC in treatments with a higher proportion of cattle dung.

## (f) Ecosystem metabolism

Dung treatment strongly influenced temporal trends in GPP and ER (electronic supplementary material §S5). There was a significant effect of dung treatment on the maximum production value (K) and the rate of increase in GPP; as the proportion of cattle dung increased, GPP increased slower but reached a higher maximum (figure 1a,b). GPP increase also started later with less cattle dung (figure 1d), but this lag effect was insignificant owing to excessively high parameter estimates for two streams (0% cattle dung) that had poor data coverage in the first two weeks. Notably, while maximum GPP increased linearly with dung treatment, the rate of increase and the lag changed nonlinearly with dung treatment and suggested stronger changes when even only

Proc. R. Soc. B 287: 20193000

**Table 1.** Results of mixed-effects models for $\log_e(X)$-transformed dissolved organic carbon (DOC, mg $l^{-1}$), chlorophyll $a$ (Chl-$a$, mg $l^{-1}$), ash-free dry mass (AFDM, mg $cm^{-2}$), total suspended solids (TSS, mg $l^{-1}$), particulate organic matter (POM, mg $l^{-1}$), soluble reactive phosphorus (SRP, µg $l^{-1}$), total phosphorus (TP, mg $l^{-1}$), ammonium (mg $l^{-1}$), nitrite (mg $l^{-1}$) and nitrate (mg $l^{-1}$). The marginal $R^2$ (GLMM(m); fixed effects only) and the conditional $R^2$ (GLMM(c); fixed and random effects) represent the proportion of variance explained by each model; s.e. = standard error; s.d. = standard deviation; $^*p < 0.05$, $^{**}p < 0.01$, $^{***}p < 0.001$.

| fixed effects | SRP β (s.e.) | ammonium β (s.e.) | nitrite β (s.e.) | nitrate β (s.e.) | DOC β (s.e.) | Chl-$a$ β (s.e.) | AFDM β (s.e.) | TSS β (s.e.) | POM β (s.e.) |
|---|---|---|---|---|---|---|---|---|---|
| intercept | 0.87 (0.08)*** | 0.1 (0.02)*** | 0.14 (0.01)*** | 0.39 (0.05)*** | 1.17 (0.03)*** | 1.11 (0.05)*** | 0.79 (0.07)*** | 1.08 (0.04)*** | 0.32 (0.10)*** |
| dung treatment | 0.004 (0.001)** | 0.001 (0.0002)* | 0.001 (0.0002)*** | 0.002 (0.001)* | 0.002 (0.001)*** | 0.003 (0.001)*** | 0.003 (0.001)* | 0.002 (0.001)* | 0.003 (0.002)* |
| time | −0.19 (0.02)*** | −0.02 (0.05)*** | −0.02 (0.003)*** | 0.03 (0.01)*** | 0.04 (0.004)*** | 0.22 (0.02)*** | 0.12 (0.02)*** | 0.11 (0.01)*** | 0.27 (0.03)*** |
| dung treatment × time | 0.001 (0.0003)* | 0.0002 (0.0001)* | 0.0002 (0.00004)*** | — | — | — | — | −0.0004 (0.0002)* | — |
| **ANOVA for fixed effects** | **F-value** | **F-value** | **F-value** | **F-value** | **F-value** | **F-value** | **F-value** | **F-value** | **F-value** |
| intercept | 58.1*** | 19.6*** | 100.0*** | 630.7*** | 7986.4*** | 4265.4*** | 1938.7*** | 6273.1*** | 857.7*** |
| dung treatment | 4.6* | 6.5** | 27.9*** | 11.6** | 21.3*** | 30.4*** | 40.0*** | 42.9*** | 33.1*** |
| time | 161.5*** | 17.4*** | 76.1*** | 43.3*** | 103.2*** | 33.7*** | 89.3*** | 285.3*** | 263.4*** |
| dung treatment × time | 6.9* | 6.3* | 20.9*** | 1.3 | 0.4 | 1.2 | 1.3 | 4.8* | 1.7 |
| **random effects** | **s.d.** | **s.d.** | **s.d.** | **s.d.** | **s.d.** | **s.d.** | **s.d.** | **s.d.** | **s.d.** |
| block (intercept) | 0.03 | <0.001 | <0.001 | <0.001 | <0.001 | <0.001 | <0.001 | <0.001 | <0.001 |
| residuals | 0.26 | 0.06 | 0.038 | 0.13 | 0.09 | 0.24 | 0.25 | 0.12 | 0.34 |
| $R^2_{GLMM(m)}$ | 0.58 | 0.20 | 0.51 | 0.34 | 0.51 | 0.74 | 0.47 | 0.72 | 0.71 |
| $R^2_{GLMM(c)}$ | 0.58 | 0.20 | 0.55 | 0.46 | 0.61 | 0.75 | 0.47 | 0.76 | 0.71 |

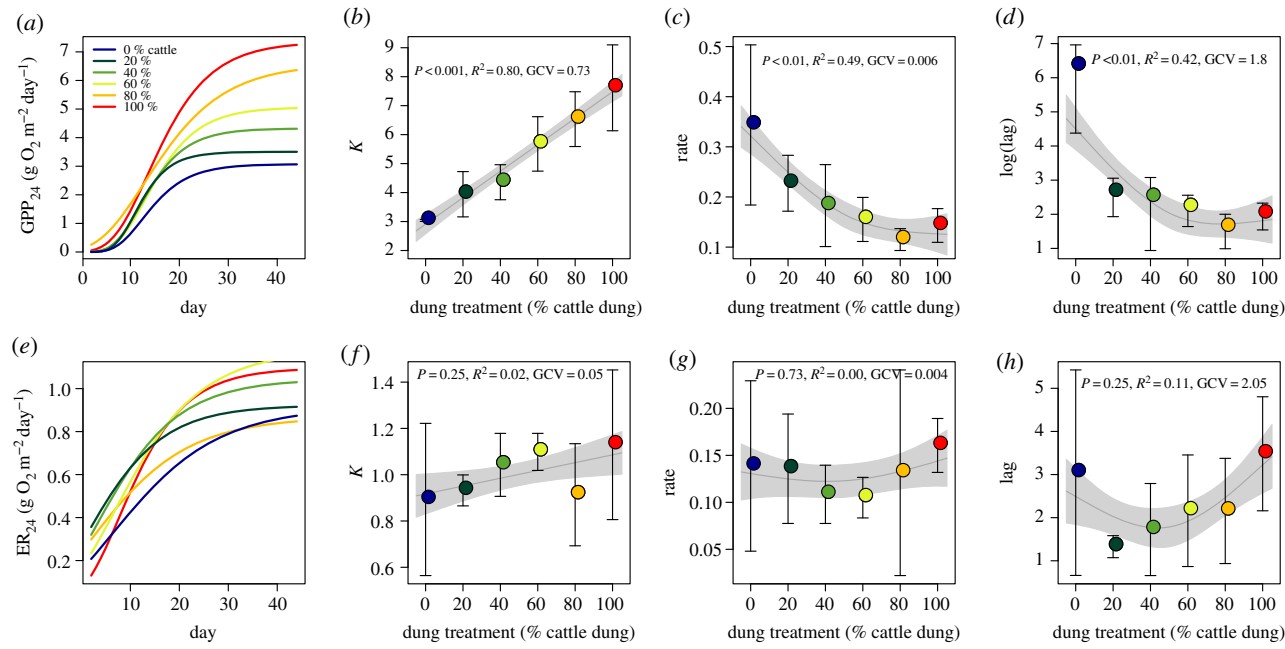

**Figure 1.** Dynamics of GPP (a) and ER (e) over 44 days as fitted with a three-parameter sigmoid Gompertz model. To test relationship with dung treatment, we plotted mean and s.d. of upper asymptote $K$, maximum rate of increase and lag for Gompertz models for GPP (b,c,d) and ER (f,g,h) as a function of dung treatment, respectively, and fitted a smoothing model (grey line with shaded area represents smoother mean and s.e.; smoother significance, $R^2$ and GCV are supplied in the figures). Note that parameter estimates for the smoothing models ($n = 3$ per treatment) were based on fits to data of individual flumes. Note also log-scale for lag in (d) owing to excessive lag in two flumes with 0% cattle dung. (Online version in colour.)

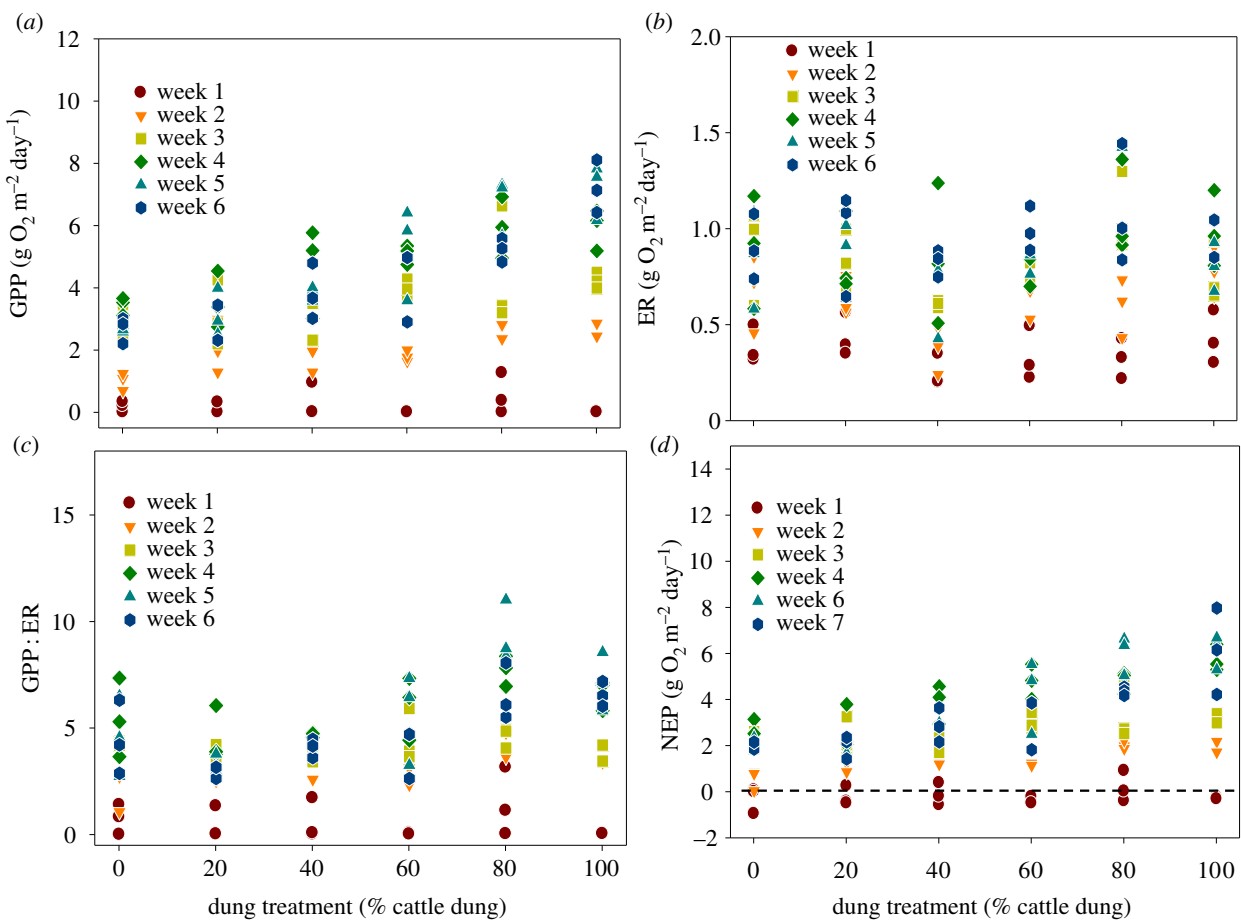

**Figure 2.** Weekly measures of flume-scale GPP (a), flume-scale ER (b), GPP : ER (c) and NEP (d). The dashed line indicates NEP = 0, and most of the mesocosms were net heterotrophic until day 7 and then switched. (Online version in colour.)

a small fraction of hippo dung was replaced by cattle dung. Because of the use of different theta values for temperature dependence of ER in metabolism models [50–52], we used a higher modelled value of 1.1085 since our attempts to use a common value of 1.045 [53] were unsuccessful. We subsequently performed a sensitivity analysis to compare results of using both theta values on the findings, and concluded that the response in GPP and ER to dung treatment

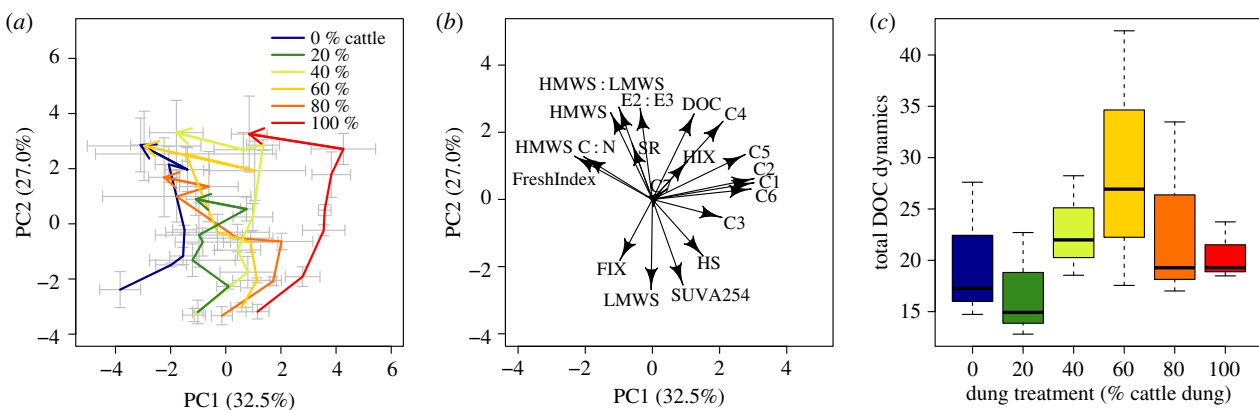

**Figure 3.** PCA based on descriptors of DOC. DOC composition changed over time towards a common endpoint composition when plotting scores (mean ± s.d. per treatment and time) (*a*). The PCA was based on PARAFAC components C1 to C7, high and low molecular weight substances (HMWS, LMWS), ratio of HMWS : LMWS and C : N of HWMS, humic-like substances (HS), aromaticity via specific ultraviolet absorption at 254 nm (SUVA), humification index (HIX), fluorescence index (FIX), freshness index $\beta : \alpha$ (FreshIndex) and an absorbance-based indicator of molecular size (E2 : E3) (*b*). Stream-specific changes of DOC composition were quantified as cumulative Euclidean distance in PCA space considering all its dimensions and progress along a path of consecutive time points; the graph shows average total path length per treatment (*c*). Note that the arrows in (*a*) designate the time series from early to late in experiment for each treatment, and 'total DOC dynamics' in (*c*) describes the approximate 'length' of the temporal arrows in (*a*). (Online version in colour.)

remained unchanged (electronic supplementary material, figures S7–S9).

Dung treatment did not significantly affect $K$, rate and lag of ER (figure 1*e–h*), though there was the suggestion of some nonlinear trends in response to dung treatment. Replacing 20–60% of hippo dung with cattle dung slightly increased the maximum rate of respiration, but the streams with 80% cattle dung had exceptionally low maximum ER (figure 1*e,f*). The rate of increase in respiration and the lag parameter showed distinctly nonlinear but insignificant trends, with lowest values at intermediate dung replacement (figure 1*g,h*). Our analysis of Gompertz parameters did not account for the blocking factor in our experimental design, but separately computed general additive mixed models did not identify a significant effect of block on any Gompertz parameter of GPP or ER (data not shown).

Using weekly averages of GPP and ER in GAMMs with a smoother for dung treatment interacting with time, and accounting for blocks as a random factor, we found a significant weekly increase in GPP with increasing proportion of cattle dung (figure 2). There was also a significant interaction between the dung treatment smoother and time (electronic supplementary material §S6, table S5), further indicating that the positive effect of cattle dung on GPP increased with time. We also found a significant main effect of dung treatment on weekly means of ER and a significant interaction between dung treatment and time (electronic supplementary material, table S5). GAMMs did not identify a significant effect of block on GPP or ER (electronic supplementary material, table S5). Most streams were heterotrophic during the first week, after which they were all autotrophic, with NEP peaking at week 4 (28 days; figure 2*a,d*).

## 4. Discussion

Our results show that replacing hippo dung with cattle dung produces different responses in aquatic ecosystems. The smaller particle sizes and higher quality (lower C : N : P ratio) of cattle dung compared with hippo dung appeared to promote increased leaching of nutrients and increased assimilation [54,55]. Indeed, cattle dung stimulated higher primary production in both the benthos and water column compared with hippo dung, although the rate of increase in GPP was nonlinear (figure 1*c*). Cattle dung also increased biofilm biomass in the water column, which is a cumulative measure of both microbial and algal production. However, there were no dung treatment effects on the fitted sigmoid parameters of ER. Hippo dung, which was composed of larger particles, tended to sink to the bottom of the streams and reduce benthic production, suggesting it may do the same in aquatic systems, especially during low flows [13,15,16]. By contrast, cattle dung tended to remain suspended or dissolve, and in aquatic systems, it may become dispersed by river discharge into a larger area, creating potentially more widespread and diffuse effects. Because light is one of the key determinants controlling production and composition of periphyton or algae in aquatic ecosystems [56], it is likely that cattle dung more strongly stimulated the autotrophic component (algae) of periphyton while hippos stimulated the heterotrophic component (bacteria/fungi), which led to higher GPP per unit biomass of periphyton among cattle dung treatments.

Although we used a theta value (1.1085) which is different from the typical value (1.045) that is commonly used in modelling GPP and ER (electronic supplementary material §S4), the conclusion reached that cattle dung stimulated higher GGP values than hippo dung remains unchanged. This is because the trends and trajectory of change in both GPP and ER are the same for both theta values (electronic supplementary material, figures S7–S9). Moreover, ER, which is more temperature sensitive than GPP [50,51], did not respond to dung treatment, irrespecive of the theta value used. This is intriguing and suggests different drivers for GPP and ER in the experiment. This can be explained by a lack of coupling between GPP and ER, which explains the increasing concentration of DOC and microbial biomass in cattle dung-dominated treatments over time. Thus, it is likely that increased ER from heterotrophs in the hippo dung treatment was offset by the increased autotrophic respiration in the cattle treatments. If there was any coupling

of GPP and ER, some variation in ER would have occurred, because a proportion of ER is autotrophic respiration [57,58].

There were also effects of dung treatment and treatment by time interactions in the composition of DOC. In streams that received higher proportions of cattle dung relative to hippo dung, DOC displayed a strong increase in diversity over time, moving from a dominance of allochthonous DOC, through a dominance of microbially produced DOC, to finally a dominance of autochthonously produced DOC from primary production (figure 3). DOC in cattle dung treatments also showed a higher contribution of humic-like components associated with microbial activity and high fractions of a fulvic acid-like component of higher plant material origin (electronic supplementary material §S6). The difference in DOC composition between hippo dung and cattle dung could be due to differences in digestion efficiency between cattle and hippos [24]; it may also result from cattle foraging on a wider selection of plants and thus encountering a wider variety of metabolites and chemicals than hippos [26,27]. The strong response of GPP to dung treatment left a strong imprint on DOC [40]. For instance, as GPP peaked over time, DOC concentration increased in concert, and composition shifted from predominantly allochthonous towards increasingly autochthonous.

While we recognize that inputs by hippos and cattle likely vary in quantity across time and space, which impacts how they affect river ecosystem function, this study specifically focused on comparing the impact of input quality from these two large herbivores. Although there are no data for African savannahs showing rates of OM and nutrient loading by livestock into rivers, preliminary findings from the Mara River show that 10–15% of cattle that visit watering points defaecate and/or urinate in the river (J. Iteba 2019, unpublished data). Our model for estimating dung inputs by cattle show that only a small fraction of daily dung production by cattle is deposited directly into the river, compared with 50% of hippo dung. However, owing to variation in cattle and hippo numbers across the landscape, cattle inputs can range from around 6 to 57% of total organic matter loading due to cattle and hippos (electronic supplementary material, table S1). Our estimates of both hippo and livestock loading have some uncertainty around them that could be improved with more detailed knowledge of animal time budgets and population sizes. For example, cattle numbers in the basin can more than double in the dry season, when livestock are herded in for increased forage [49], suggesting our estimates for cattle loading are conservative. Although the majority of cattle dung is deposited outside the river, some proportion of it likely enters aquatic systems during large rainfall events. Furthermore, it is likely that the trend towards increasing populations of cattle and other livestock, such as goats and sheep, is likely to continue.

This research increases our knowledge about how resource subsidies from cattle may influence aquatic ecosystem function and highlights similarities and differences between subsidies transported by cattle versus hippos. Similar to other LMH such as hippos and ungulates [9,12], cattle can create biogeochemical hotspots through congregation and egestion. However, cattle subsidies are more likely to increase nutrient concentrations and stimulate primary production in recipient aquatic ecosystems, which could have pronounced bottom-up effects on food webs. In addition, other aspects of cattle behaviour may have pronounced ecosystem effects. The development of cattle footpaths can channel water and nutrients from terrestrial to aquatic environments. Large herds of cattle visit watering points during the dry season and concentrate in riparian areas, where they deposit dung and urine, which may contribute substantially to nutrient flux at a time when low water runoff limits fluxes by hydrological vectors. Dung and urine deposited around waterholes also enrich the soil and vegetation, and this enrichment could increase fluxes of organic matter for aquatic consumers during inundation and/or litterfall.

## 5. Conclusion

Here we show that cattle and hippo dung have contrasting effects on aquatic ecosystem function, likely caused by differences in faecal particle size and stoichiometry of major elements (C : N : P ratio). Increasing inputs of cattle dung led to higher GPP and a more complex and diverse DOC composition. By contrast, hippo dung reduced benthic primary production and led to a delayed response in GPP, which is consistent with whole-river observations [16,59]. In landscapes where livestock are displacing hippos, these differences may lead to substantial changes in aquatic ecosystem structure and function. Taken collectively, our results expand the current understanding of the role played by large mammalian herbivores in the functioning of aquatic ecosystems in African savannahs. However, they also emphasize the species-specific nature of many of these ecological roles and suggest that species introductions and/or rewilding efforts seeking to replace extinct species with modern analogues may have unintended outcomes [60,61]. Our results highlight the need for more research on the ecological consequences of introduced large herbivores and replacement of native populations by anthropogenic change.

Data accessibility. The data supporting this article have been deposited with the Dryad Digital Repository at https://doi.org/10.5061/dryad.jh9w0vt79 [62].

Authors' contributions. F.O.M. conceived of the study, designed the study, collected field data, performed data analysis and drafted the manuscript. M.J.K. and C.R.G.-Q. collected field data and performed laboratory sample analysis; A.L.S., C.L.D. and D.M.P. designed the study and critically revised the manuscript; G.A.S. designed the study, performed data analysis and critically revised the manuscript. All authors gave final approval for publication and agree to be held accountable for the work performed herein.

Competing interests. We declare we have no competing interests.

Funding. This work was supported by the International Foundation for Science (Research grant no. A/5810-1), an Alexander von Humboldt Postdoc fellowship to F.O.M.; the German research foundation (within the Research Training Group on Urban Water Interfaces, GRK 2032) to C.R.G.-Q. and the U.S. National Science Foundation to D.M.P. (DEB 1354053 and DEB 1753727).

Acknowledgements. We are grateful to Gilbert Geemi and Paul Geemi for assistance during installation of experimental mesocosms and field sampling. We acknowledge the Mara River Water User's Association for hosting our array of mesocosms. We are grateful to Tobias Goldhammer, Sarah Krocker and Claudia Schmalsch for help during analysis at the Leibniz Institute of Freshwater Ecology and Inland Fisheries (IGB). We acknowledge Sylvia Ortmann (Leibniz Institute for Zoo and Wildlife Research) for chemical analysis of hippo and cattle dung. Emma J. Rosi (Cary Institute of Ecosystem Studies, NY) provided many helpful comments on the manuscript.

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
