## [Reviewer comments · Proceedings of the Royal Society B: Biological Sciences]

Review History

RSPB-2019-0650.R0 (Original submission)

Review form: Reviewer 1

Recommendation

Accept with minor revision (please list in comments)

Scientific importance: Is the manuscript an original and important contribution to its field?

Excellent

General interest: Is the paper of sufficient general interest?

Excellent

Quality of the paper: Is the overall quality of the paper suitable?

Excellent

Is the length of the paper justified?

Yes

Should the paper be seen by a specialist statistical reviewer?

No

Do you have any concerns about statistical analyses in this paper? If so, please specify them explicitly in your report.

Yes

It is a condition of publication that authors make their supporting data, code and materials available - either as supplementary material or hosted in an external repository. Please rate, if applicable, the supporting data on the following criteria.

Is it accessible?

N/A

Is it clear?

N/A

Is it adequate?

N/A

Do you have any ethical concerns with this paper?

No

Comments to the Author

I have now read the manuscript by Masese et al. The manuscript experimentally assesses how terrestrial subsidies to aquatic ecosystems vary in quality along a gradient from 100% hippo dung to 100% cattle dung, as well as their effects on nutrients, organic matter, and ecosystem metabolism. They found strong differences of dung treatment on dissolved nutrients, organic matter, biomass of biofilms, chlorophyll, ecosystem metabolism and DOM composition. Gross primary production was increased by cattle dung, likely due to its higher quality (higher C:N ratio), and smaller particle size; in contrast, hippo dung tended to sink, and decreased primary production. Curiously enough, these effects on primary production were not mirrored by changes in ecosystem respiration. These results are relevant in the current context of increasing densities of cattle in many regions where wild mammalian herbivores used to be the main agent subsidising aquatic ecosystems. It is also relevant to further prove that not all mammalian herbivores influence the same ecosystem processes in the context of rewilding.

I think the authors have done a great job in summarising a huge amount of results of different nature (from dissolved nutrients, to DOM quality, to ecosystem metabolism), and analysed with very different techniques, which has produced a manuscript that is straightforward, easy-to-read and relevant for a general audience. I do have some concerns, of course, that I will specify below, but none of them is of a general nature – they are very specific issues that should not be difficult to address in a revised version. I therefore recommend the acceptance of this manuscript with minor revision.

SPECIFIC COMMENTS

Abstract is great. Straightforward and very easy to follow.

Introduction is perfect. I would not change a thing.

Materials and methods.

Lines 128-131. It is a bit difficult to imagine the experimental design without Figure S1. I would therefore suggest moving it from the supplementary to the main manuscript (if space allows).

Line 192. I think this is the first time 'DO' appears. It should be defined (I guess it's Dissolved Oxygen?).

Line 197 and Supplementary Information 2. According to the authors, the temperature dependency of ecosystem respiration (θ , θ) is a 'tricky' parameter in metabolic models. The

authors state that it is quite an important parameter, and that there has been a lot of discussion around it. In this sense, it would therefore be useful to conduct a sensitivity analysis of this parameter maybe? Or at least to see the effects of tuning it on the shapes of the plots in Figure 2. E.g. how do the plots in fig 2 look like if the authors use the 'commonly used value of 1.045 instead of 1.1085? How they ended up with this final value of 1.1085? How did you assess the goodness of fit of the models? In other words, what does it mean the sentence 'our models were better constrained [with higher theta]'.

Line 199. It appears to me that you used Linear Mixed Effects models instead of GLMMs. I say this because you said you used the function lme() (line 200) and because you say you transformed some variables to meet normality assumptions (lines 205-6).

Line 211. Can you please explain a little bit to the non-specialist reader what each of these parameters mean?

Line 212. Why did you pool data from 3 streams? Is that the reason why you don't get error bands in Fig 2a,e? But then why are there confidence intervals and error bars for the rest of the panels in Fig 2? Because these are the error bands coming from regression? This bit needs a bit of clarification, since you state you used both the data pooled and the data for each stream, but I'm not sure how was done, in the end (I would say pooled).

Results.

Line 234. 'Cow' should be replaced by 'cattle'.

Line 245. 'Figure S2a,b' should be 'Figure S2a,c', I think.

Line 246. 'Figure S2c,d' should be 'Figure S2b,d', I think.

Line 275-276. Was this because you pooled data from the 3 replicate streams?

Line 283-284. '... further indicating that the positive effect of cattle dung on GPP increased with time.' Where can we see these results? Is it in Supplementary information 4, Table S2?

Table 1. These appear to be the results from doing summary(model) in R. These tests, check the significance of the parameters (whether their estimate cross the 0 line). To check the main effects, you should also conduct the test of main effects, using anova() or Anova() in R. Can you report these results?

Lines 290-296 need a bit of editing. Currently I find them difficult to follow.

Line 292. I cannot see the non-linear trend (line 290) in Figure 2.

Line 293. It should be Supporting Information 4, not 5, I think.

Figure 1. The labels (a-h) for each panel are lacking.

Line 316. Supporting Information 5, not 6, I think.

Line 324-326. Where can I see the results described here?

Lines 326-328. Where can I see these results?

Lines 330-331. Where can I see these results?

Line 338. After '...cattle were slightly lower again' should go (Figure 3c), I think.

You should also describe Figure 3b. Currently not described in the results section.

Discussion.

Line 362. Replace 'accumulates' for 'accumulate'.

Line 367. 'Replace' increasing by 'increased'.

Line 369. Figure 1b, should be Figure 1c.

Line 371. Figure 1b, should be Figure 1c.

Lines 374-384. This paragraph is great, but it's very difficult for the general readers to jump from the results section to these conclusions if you don't give them a bit more context (either in the results, or here, or supplementary).

Line 378. Figure 3 should be referenced here, instead of Figure 2, I think.

Lines 428. '... reduced benthic primary production...' compared to the treatments with high % of cattle dung treatments. What about compared to water without 'subsidies'? You didn't have this treatment in this study, but it would be interesting to discuss somewhere in the discussion what is the overall effect of large herbivorous mammal subsidies in streams. And then talk about how domestic vs. wild large herbivores influence differently ecosystem processes, which is the focus of this study.

Review form: Reviewer 2

Recommendation

Accept with minor revision (please list in comments)

Scientific importance: Is the manuscript an original and important contribution to its field?

Excellent

General interest: Is the paper of sufficient general interest?

Good

Quality of the paper: Is the overall quality of the paper suitable?

Excellent

Is the length of the paper justified?

Yes

Should the paper be seen by a specialist statistical reviewer?

No

Do you have any concerns about statistical analyses in this paper? If so, please specify them explicitly in your report.

No

It is a condition of publication that authors make their supporting data, code and materials available - either as supplementary material or hosted in an external repository. Please rate, if applicable, the supporting data on the following criteria.

Is it accessible?

Yes

Is it clear?

Yes

Is it adequate?

Yes

Do you have any ethical concerns with this paper?

No

Comments to the Author

Reviewer comments for Masese et al. submitted to Proceedings B, "Wild mammalian herbivores are distinct from domestic livestock in their resource subsidies to and effects on aquatic ecosystems"

This manuscript describes results of a study comparing the effects of hippo versus cattle dung on recipient aquatic ecosystems. The study used experimental mesocosms amended with a wide range of combinations of hippo and cattle dung, establishing a gradient to investigate interactive and singular effects of each on water chemistry, periphyton, ecosystem metabolism, and dissolved organic matter. The study shows that nutrient-rich cattle dung drive greater autotrophy within recipient systems, with implications for nutrient dynamics, periphyton growth, gross primary production, and the chemistry of dissolved organic matter. The intensive characterization, extended duration, and wide treatment gradient are real strengths of the study - as are the clear implications for aquatic ecosystems under global change. I do have recommendations for revision of the manuscript, below.

Major comments:

The experimental design is a nice test of the effects of cattle vs. hippo dung on aquatic ecosystems, but I would have liked to see an experimental control treatment (no dung added) or a gradient of absolute quantity added, instead of different quantities of each in combination. This would provide a picture of a broader question of how dung affects recipient systems, and how much the effects of cattle dung specifically compare to effects of simply adding any dung. Is there a reason you did not include a control treatment, and would you address this in your discussion? Based on the title and nature of this study, I was expecting to see more data quantifying the inputs of material (at least dung) from cattle versus hippos measured in the field. This seems like a major remaining knowledge gap, because it is not clear whether the experimental conditions in your study mimic the actual mode and quantity of inputs to aquatic ecosystems. The paragraph of lines 390-403 addresses these concerns well, but the authors should perhaps be more specific about what questions remain a priority for future studies, either resulting from their own findings...or to put the results in a more realistic context.

Line comments:

Line 53 – Please break up this sentence by adding commas such as after grasslands, forages, and ecosystems.

Lines 79-81 – Nice detail showing the contrasting C:N:P of dung from different large herbivores, but please qualify this sentence because the writing makes it seem these ratios are fixed, but dung C:N:P surely vary over time and across individuals from each species. For example, consider listing mean +/- SE or simply mention these data were measured from a subset of individuals.

Line 200 – NO₃ has only 1 minus in its superscript

Line 214 – I am not very familiar with the term “lag” for describing temporal changes in metabolism. Can you explain here what this term describes – for example, what do high versus low values of “lag” indicate?

Line 239 – Please cite Table S1 at the end of this sentence.

Lines 241-246 – These temporal changes in nutrient concentrations are very interesting, but are difficult to interpret in the supplemental figures because the legend is time. Please consider making time the X-axis in these figures, as you have done in Figure 1a, to show the temporal dynamics across treatments.

Line 362 – accumulate

Line 366 – I agree that the effects of cattle dung may be more widespread, but the high capacity for transport suggests that these effects will also be more diffuse (not as high of magnitude or persistent at the site of subsidy, for example).

Lines 367-373 – This paragraph is short and feels under-developed. I would be interested to see more discussion of what the contrasting GPP vs. ER responses to cattle dung might mean. Do cattle dung more strongly stimulate algae than they stimulate heterotrophic components (bacteria/fungi) of the periphyton, for example? Perhaps cattle dung not only stimulate algal growth...but also increase algal health, such that there is more GPP per unit periphyton chlorophyll or ER, and the experiment did not run long enough for algal mortality to ensue and stimulate respiration?

Lines 385-389 – This paragraph also seems rather short. Some of this material could perhaps be combined with the previous 2 paragraphs. I find it quite interesting that potential DOM sources shift depending on the dung type, not only at the beginning of the study but also long-term (later in the study) by indirectly affecting autotrophy.

Line 417 – unclear what “their” refers to. Cattle dung?

Lines 431-433 – This final sentence seemed out-of-place for the paper, because rewilding is not mentioned elsewhere in the paper, and there is not a clear connection to your specific study findings. For example, are cattle being used in rewilding efforts...or are hippos receiving more protections toward rewilding? Consider elaborating on these specifics in the introduction, or instead, focusing on broad contrasts between wildlife and livestock as mediators of subsidies from terrestrial to aquatic systems, e.g. returning to ecological implications of points raised in lines 60-73. Based on your experimental findings, how might replacement of wild herbivores by domesticated livestock change aquatic ecosystems...or are the 2 groups functionally quite similar?

Figure 1 – Panel letters are missing from the figure proper. In (b), (c), (f), and (g) – K and Rate

have units of g O₂ per day? Are values per unit area (meter squared), as in Figure 2? Figure 3 – I really like the arrows in panel (a). Consider adding a note in the figure title that the arrows designate the time series from early to late in experiment for each treatment. My interpretation is that the response variable “total DOM dynamics” in (c) describes the approximate ‘length’ of the temporal arrows in panel (a) – can a note to this effect also be added?

Decision letter (RSPB-2019-0650.R0)

22-Jul-2019

Dear Dr Masese:

I am writing to inform you that your manuscript RSPB-2019-0650 entitled "Wild mammalian herbivores are distinct from domestic livestock in their resource subsidies to and effects on aquatic ecosystems" has, in its current form, been rejected for publication in Proceedings B.

This action has been taken on the advice of referees and the Associate Editor, who have recommended that substantial revisions are necessary. With this in mind we are willing to consider a resubmission, provided the comments of the referees and the Associate Editor are fully addressed. It is important to note that this is not a provisional acceptance.

- 1) A ‘response to referees’ document including details of how you have responded to the comments, and the adjustments you have made.
- 2) A clean copy of the manuscript and one with 'tracked changes' indicating your 'response to referees' comments document.
- 3) Line numbers in your main document.

Sincerely,
Victoria Braithwaite

=====
Professor V A Braithwaite
mailto: proceedingsb@royalsociety.org
=====

Associate Editor, Comments to Author:

We have now received two reviews of this paper on this manuscript on the subsidies by cows and hippos to aquatic ecosystems. Both reviewers found a great deal of potential in this

manuscript, and I agree. Unfortunately there very serious conceptual points that are not well dealt with in the current manuscript.

The title doesn't reflect the paper very well. First, it creates the impression that this will be a general review of all wild mammalian herbivores, which the paper is definitely not. Hippos are, of course, an example of mammalian herbivores, but the reader cannot and should not extrapolate from hippos to other mammalian herbivores based on the evidence in this paper. In theory, the authors could build this argument in the discussion, perhaps based on C:N:P ratios of other wild mammalian herbivores. In fact the evidence cited about the East African Buffalo works against the authors own argument. (In this section the authors also mis-use the term "foregut fermenters", which is a group includes ruminants and hippos, ie ruminants definitely ferment in their foreguts.)

The other major conceptual issue raised by reviewer 2 is about mass of added organic material. This is a very serious shortcoming, especially given recent evidence about the effects of hippo dung published by one of the co-authors (Dutton et al. 2018). It seems likely that per unit mass of forage hippos are producing more mass of dung than cattle and that sheer quantity of organic matter is having an effect on aquatic ecosystems, an effect that is not captured by the authors' experiments.

Together these points suggest that main conceptual result in this manuscript is about the effect of ruminants versus pseudo-ruminants per unit feces, which is interesting, but it's not globally applicable in the way that a general comparison of wild mammals and livestock would be. (This point is also made by reviewer 1.)

Certainly the ecosystem level effects of cattle and hippos are important and interesting, but there is a serious limitation to how much a per unit feces experiment tells us. My recommendation is two-fold: first, the title and some of the framing of the paper needs to change to reflect that it's not reasonable to extrapolate to all wild mammalian herbivores. Second, these results need to be embedded in an ecosystem model to actually be informative about the effect of these two species on ecosystems. This model does not have to be complicated, in fact it's better if it's not, but there is certainly reason to believe that the per unit feces results might be out-weighted by difference in the total amount of subsidy. I realize this is a difficult request, but otherwise the results of this experiment alone are not enough to understand the effect of these two species on the ecosystem.

Dutton, C. L., Subalusky, A. L., Hamilton, S. K., Rosi, E. J., & Post, D. M. (2018). Organic matter loading by hippopotami causes subsidy overload resulting in downstream hypoxia and fish kills. *Nature communications*, 9(1), 1951.

====

Reviewers' Comments to Author:

Referee: 1

I have now read the manuscript by Masese et al. The manuscript experimentally assesses how terrestrial subsidies to aquatic ecosystems vary in quality along a gradient from 100% hippo dung to 100% cattle dung, as well as their effects on nutrients, organic matter, and ecosystem metabolism. They found strong differences of dung treatment on dissolved nutrients, organic matter, biomass of biofilms, chlorophyll, ecosystem metabolism and DOM composition. Gross primary production was increased by cattle dung, likely due to its higher quality (higher C:N ratio), and smaller particle size; in contrast, hippo dung tended to sink, and decreased primary production. Curiously enough, these effects on primary production were not mirrored by changes in ecosystem respiration. These results are relevant in the current context of increasing densities of cattle in many regions where wild mammalian herbivores used to be the main agent

subsidising aquatic ecosystems. It is also relevant to further prove that not all mammalian herbivores influence the same ecosystem processes in the context of rewilding.

I think the authors have done a great job in summarising a huge amount of results of different nature (from dissolved nutrients, to DOM quality, to ecosystem metabolism), and analysed with very different techniques, which has produced a manuscript that is straightforward, easy-to-read and relevant for a general audience. I do have some concerns, of course, that I will specify below, but none of them is of a general nature – they are very specific issues that should not be difficult to address in a revised version. I therefore recommend the acceptance of this manuscript with minor revision.

SPECIFIC COMMENTS

Abstract is great. Straightforward and very easy to follow.

Introduction is perfect. I would not change a thing.

Materials and methods.

Lines 128-131. It is a bit difficult to imagine the experimental design without Figure S1. I would therefore suggest moving it from the supplementary to the main manuscript (if space allows).

Line 192. I think this is the first time 'DO' appears. It should be defined (I guess it's Dissolved Oxygen?).

Line 197 and Supplementary Information 2. According to the authors, the temperature dependency of ecosystem respiration (θ , θ) is a 'tricky' parameter in metabolic models. The authors state that it is quite an important parameter, and that there has been a lot of discussion around it. In this sense, it would therefore be useful to conduct a sensitivity analysis of this parameter maybe? Or at least to see the effects of tuning it on the shapes of the plots in Figure 2. E.g. how do the plots in fig 2 look like if the authors use the 'commonly used value of 1.045 instead of 1.1085? How they ended up with this final value of 1.1085? How did you assess the goodness of fit of the models? In other words, what does it mean the sentence 'our models were better constrained [with higher theta]'.

Line 199. It appears to me that you used Linear Mixed Effects models instead of GLMMs. I say this because you said you used the function `lme()` (line 200) and because you say you transformed some variables to meet normality assumptions (lines 205-6).

Line 211. Can you please explain a little bit to the non-specialist reader what each of these parameters mean?

Line 212. Why did you pool data from 3 streams? Is that the reason why you don't get error bands in Fig 2a,e? But then why are there confidence intervals and error bars for the rest of the panels in Fig 2? Because these are the error bands coming from regression? This bit needs a bit of clarification, since you state you used both the data pooled and the data for each stream, but I'm not sure how was done, in the end (I would say pooled).

Results.

Line 234. 'Cow' should be replaced by 'cattle'.

Line 245. 'Figure S2a,b' should be 'Figure S2a,c', I think.

Line 246. 'Figure S2c,d' should be 'Figure S2b,d', I think.

Line 275-276. Was this because you pooled data from the 3 replicate streams?

Line 283-284. '... further indicating that the positive effect of cattle dung on GPP increased with time.' Where can we see these results? Is it in Supplementary information 4, Table S2?

Table 1. These appear to be the results from doing summary(model) in R. These tests, check the significance of the parameters (whether their estimate cross the 0 line). To check the main effects, you should also conduct the test of main effects, using `anova()` or `Anova()` in R. Can you report these results?

Lines 290-296 need a bit of editing. Currently I find them difficult to follow.

Line 292. I cannot see the non-linear trend (line 290) in Figure 2.

Line 293. It should be Supporting Information 4, not 5, I think.

Figure 1. The labels (a-h) for each panel are lacking.

Line 316. Supporting Information 5, not 6, I think.

Line 324-326. Where can I see the results described here?

Lines 326-328. Where can I see these results?

Lines 330-331. Where can I see these results?

Line 338. After ‘...cattle were slightly lower again’ should go (Figure 3c), I think.

You should also describe Figure 3b. Currently not described in the results section.

Discussion.

Line 362. Replace ‘accumulates’ for ‘accumulate’.

Line 367. ‘Replace’ increasing by ‘increased’.

Line 369. Figure 1b, should be Figure 1c.

Line 371. Figure 1b, should be Figure 1c.

Lines 374-384. This paragraph is great, but it’s very difficult for the general readers to jump from the results section to these conclusions if you don’t give them a bit more context (either in the results, or here, or supplementary).

Line 378. Figure 3 should be referenced here, instead of Figure 2, I think.

Lines 428. ‘... reduced benthic primary production...’ compared to the treatments with high % of cattle dung treatments. What about compared to water without ‘subsidies’? You didn’t have this treatment in this study, but it would be interesting to discuss somewhere in the discussion what is the overall effect of large herbivorous mammal subsidies in streams. And then talk about how domestic vs. wild large herbivores influence differently ecosystem processes, which is the focus of this study.

Referee: 2

Reviewer comments for Masese et al. submitted to Proceedings B, “Wild mammalian herbivores are distinct from domestic livestock in their resource subsidies to and effects on aquatic ecosystems”

This manuscript describes results of a study comparing the effects of hippo versus cattle dung on recipient aquatic ecosystems. The study used experimental mesocosms amended with a wide range of combinations of hippo and cattle dung, establishing a gradient to investigate interactive and singular effects of each on water chemistry, periphyton, ecosystem metabolism, and dissolved organic matter. The study shows that nutrient-rich cattle dung drive greater autotrophy within recipient systems, with implications for nutrient dynamics, periphyton growth, gross primary production, and the chemistry of dissolved organic matter. The intensive characterization, extended duration, and wide treatment gradient are real strengths of the study – as are the clear implications for aquatic ecosystems under global change. I do have recommendations for revision of the manuscript, below.

Major comments:

The experimental design is a nice test of the effects of cattle vs. hippo dung on aquatic ecosystems, but I would have liked to see an experimental control treatment (no dung added) or a gradient of absolute quantity added, instead of different quantities of each in combination. This would provide a picture of a broader question of how dung affects recipient systems, and how much the effects of cattle dung specifically compare to effects of simply adding any dung. Is there a reason you did not include a control treatment, and would you address this in your discussion? Based on the title and nature of this study, I was expecting to see more data quantifying the inputs of material (at least dung) from cattle versus hippos measured in the field. This seems like a major remaining knowledge gap, because it is not clear whether the experimental conditions in your study mimic the actual mode and quantity of inputs to aquatic ecosystems. The paragraph of lines 390-403 addresses these concerns well, but the authors should perhaps be more specific about what questions remain a priority for future studies, either resulting from their own findings...or to put the results in a more realistic context.

Line comments:

Line 53 – Please break up this sentence by adding commas such as after grasslands, forages, and ecosystems.

Lines 79-81 – Nice detail showing the contrasting C:N:P of dung from different large herbivores, but please qualify this sentence because the writing makes it seem these ratios are fixed, but dung C:N:P surely vary over time and across individuals from each species. For example, consider listing mean +/- SE or simply mention these data were measured from a subset of individuals.

Line 200 – NO₃ has only 1 minus in its superscript

Line 214 – I am not very familiar with the term “lag” for describing temporal changes in metabolism. Can you explain here what this term describes – for example, what do high versus low values of “lag” indicate?

Line 239 – Please cite Table S1 at the end of this sentence.

Lines 241-246 – These temporal changes in nutrient concentrations are very interesting, but are difficult to interpret in the supplemental figures because the legend is time. Please consider making time the X-axis in these figures, as you have done in Figure 1a, to show the temporal dynamics across treatments.

Line 362 – accumulate

Line 366 – I agree that the effects of cattle dung may be more widespread, but the high capacity for transport suggests that these effects will also be more diffuse (not as high of magnitude or persistent at the site of subsidy, for example).

Lines 367-373 – This paragraph is short and feels under-developed. I would be interested to see more discussion of what the contrasting GPP vs. ER responses to cattle dung might mean. Do cattle dung more strongly stimulate algae than they stimulate heterotrophic components (bacteria/fungi) of the periphyton, for example? Perhaps cattle dung not only stimulate algal growth...but also increase algal health, such that there is more GPP per unit periphyton chlorophyll or ER, and the experiment did not run long enough for algal mortality to ensue and stimulate respiration?

Lines 385-389 – This paragraph also seems rather short. Some of this material could perhaps be combined with the previous 2 paragraphs. I find it quite interesting that potential DOM sources shift depending on the dung type, not only at the beginning of the study but also long-term (later in the study) by indirectly affecting autotrophy.

Line 417 – unclear what “their” refers to. Cattle dung?

Lines 431-433 – This final sentence seemed out-of-place for the paper, because rewilding is not mentioned elsewhere in the paper, and there is not a clear connection to your specific study findings. For example, are cattle being used in rewilding efforts...or are hippos receiving more protections toward rewilding? Consider elaborating on these specifics in the introduction, or instead, focusing on broad contrasts between wildlife and livestock as mediators of subsidies from terrestrial to aquatic systems, e.g. returning to ecological implications of points raised in lines 60-73. Based on your experimental findings, how might replacement of wild herbivores by domesticated livestock change aquatic ecosystems...or are the 2 groups functionally quite similar?

Figure 1 – Panel letters are missing from the figure proper. In (b), (c), (f), and (g) – K and Rate have units of g O₂ per day? Are values per unit area (meter squared), as in Figure 2?

Figure 3 – I really like the arrows in panel (a). Consider adding a note in the figure title that the arrows designate the time series from early to late in experiment for each treatment. My interpretation is that the response variable “total DOM dynamics” in (c) describes the approximate ‘length’ of the temporal arrows in panel (a) – can a note to this effect also be added?

Author's Response to Decision Letter for (RSPB-2019-0650.R0)

See Appendix A.

RSPB-2019-3000.R0

Review form: Reviewer 1

Recommendation

Accept with minor revision (please list in comments)

Scientific importance: Is the manuscript an original and important contribution to its field?

Excellent

General interest: Is the paper of sufficient general interest?

Excellent

Quality of the paper: Is the overall quality of the paper suitable?

Excellent

Is the length of the paper justified?

Yes

Should the paper be seen by a specialist statistical reviewer?

No

Do you have any concerns about statistical analyses in this paper? If so, please specify them explicitly in your report.

No

It is a condition of publication that authors make their supporting data, code and materials available - either as supplementary material or hosted in an external repository. Please rate, if applicable, the supporting data on the following criteria.

Is it accessible?

No

Is it clear?

N/A

Is it adequate?

N/A

Do you have any ethical concerns with this paper?

No

Comments to the Author

General comment:

I have now read the manuscript by Masese et al. that was resubmitted to Proceedings B. I found the manuscript well-written, straightforward and relevant for a general audience already in the first round, and this hasn't changed. The flow of ideas is again very nice, and the study combines many different perspectives to arrive to the conclusion that cattle subsidies to aquatic ecosystems are fundamentally different than hippo loadings. This has consequences for ecosystems that are increasingly being influenced by cattle grazing. Nearly all of the issues I highlighted on the first round of revision have been addressed. Some bits of the manuscript have been toned down a bit (e.g. the title), but I don't think these changes detract any relevance or interest. I will specify some areas of improvement below, but all of my comments are very specific. The manuscript is of high quality in general. I would recommend acceptance with minor revisions.

Specific comments:

Abstract

- There is quite a lot of emphasis to the new estimates of cattle vs. hippo loadings. While these estimates are interesting, in the sense that they set a context to assess the relevance of this manuscript's results, they are 'big numbers' (see other comments below). Therefore, I would recommend giving them less emphasis in the abstract.

Introduction

- Lines 74-77. You say that 'Ruminants such as cattle [...] have a relatively efficient digestive system compared to non-ruminants...', then you also say that 'non-ruminants, such as hippos, have longer mean retention times than ruminants...'. However, in the Supplementary 1(a), you say: 'both cattle and hippos have long mean gut retention times'. Which of these sentences is the correct one? And importantly, if hippo and cattle gut retention times are different, as I would imagine, did you model cattle loadings using the gut retention times of hippos, as I gather from reading Supplementary 1(a) and Subalusky et al 2015 Freshwater Biology paper?

- Apart from the need to clarify the previous point, the introduction is great.

Methods

- Line 129. A reference to support this method to estimate protein would be useful here.

- Lines 136-137. I would recommend you to make it very clear to the reader that in this paper you only modelled cattle loadings, because hippo loadings had already been calculated in Subalusky et al 2015. Otherwise you end up a bit lost trying to find out how you calculated hippo loadings. E.g. '... and compared results with EXISTING estimateS of loading rates for hippos in the river EXTRACTED FROM REF. [12]'

- It would be important to acknowledge somewhere that the estimates of the loading rates for cattle and hippos are quite broad or 'crude'. E.g. hippo data comes from captive hippos, the authors assume the same movement patterns for cattle during wet and dry seasons, and between protected (with predators) and unprotected (with less predators, I guess) areas...

- Line 186. You don't need to apologise for not using GAMMs, hehe, especially if you fulfil the requirements to apply a less complex and 'more parsimonious' method (linear mixed effects models). I would just say that you used LMEMs and that you fulfilled normality and homoscedasticity.

- Line 195. Introduce why you want to fit the sigmoid Gompertz. E.g. 'To test the effect of cattle dung on ecosystem respiration, or production or...'. Otherwise the paragraph appears a bit out of the blue.

Results

- Line 222. Try to use the past for all results. Here, I would write: 'Cattle dung haD remarkably...'

- Lines 235-236. Can you clarify these results? There's an question mark that appears out of place. I think this paragraph is an interesting addition, although I believe these estimates are quite grosso modo.

- Lines 252-253. If these variables increased with dung %, the coefficients in table 1 for dung should be positive. I see in the plots (FigS4) that the lines do increase, so the negative sign (Table1) must be an error, I guess.

- Line 266 and Fig. 1c. I think there has been a misunderstanding here. According to what I see in Fig. 1a, the effect of cattle dung was to increase K and ALSO INCREASE the rate at which K is achieved. However, the text and Fig 1c don't agree with this. I think the issue is that according to Supplementary Information 4, page 11, after equation (2), we can read: 'P2 (m² min⁻¹ g⁻¹ O₂) is the inverse maximum photosynthesis rate'. Therefore, if the rate plotted in Fig.1c, is the inverse of the rate at which K is achieved, everything makes sense again. And the text should be amended as follows: '... GPP increased FASTER AND reached a higher maximum (Fig. 1ab)'.

- Line 267. Reference to Fig. 1c should be FIG.1D.

- Lines 266-269. I don't think you need to justify this result, because I really can't see differences in the lag.

- Figure 2. Why didn't you plot lines on figure 2? (as done in figures S2 or S4).

- Line 288. Can you explain how you get to the conclusion that blocks were not important, based on Table S4. I don't know if the values present in the line starting Block (EDF(F)) are the p-values, or not...
- Figure legend of Fig. 1. After the last word, 'cattle' add 'dung'.
- Figure legend on Fig. 2. Line 308, delete 'on'.

Discussion

- Line 355. Here you say it correctly according to what can be seen in Fig 1a (see previous comment about the rate in plot Fig.1c, being the inverse of the rate, in fact).
- Line 356. I would write biofilm/periphyton biomass to make things easier to those that haven't read the methods.
- Lines 361-367. You don't really explain why you didn't find increased respiration in 0% cattle dung (thus 100% hippo dung) treatments... as it would be expected and as hypothesised in the introduction.
- Lines 369-372. Without a reference to Supplementary Information 6, these conclusions are quite a leap from the results. I understand the space constraints, but it really means the average reader has to 'have faith' in what you say, unless they go and read the supplementary in depth.
- Nice discussion and conclusions!

Review form: Reviewer 3

Recommendation

Major revision is needed (please make suggestions in comments)

Scientific importance: Is the manuscript an original and important contribution to its field?

Good

General interest: Is the paper of sufficient general interest?

Good

Quality of the paper: Is the overall quality of the paper suitable?

Good

Is the length of the paper justified?

Yes

Should the paper be seen by a specialist statistical reviewer?

No

Do you have any concerns about statistical analyses in this paper? If so, please specify them explicitly in your report.

Yes

It is a condition of publication that authors make their supporting data, code and materials available - either as supplementary material or hosted in an external repository. Please rate, if applicable, the supporting data on the following criteria.

Is it accessible?

Yes

Is it clear?

Yes

Is it adequate?

Yes

Do you have any ethical concerns with this paper?

No

Comments to the Author

This study presents results from a mesocosm experiment aimed at quantifying ecosystem responses to varying qualities of mammalian dung. The study design maintains a similar amount of dung in mesocosms, but varies the contribution of cattle vs. hippo. Each week, various ecosystem structural and functional components are measured and/or modeled. The authors find that cattle dung is a higher quality subsidy that positively influences nutrients and biomass, and cattle dung changes the character and temporal trajectory of DOC. The authors conclude that changes in LMH identity and numbers may thus have a large influence on river ecosystems, such as the Mara. The authors also construct a simple model to estimate total dung loading by cattle vs. hippos.

Overall, the study is well conceived and executed, and complements a number of other studies in the system – largely focused on hippos. The analysis appears comprehensive, but at times comes across as a bit overcomplicated or overparameterized – especially in the context of the temporal dynamics and magnitude of ecosystem metabolism. I do think this is an important contribution that should be of interest to many ecologists – and hope that my comments help to strengthen the presentation.

The data presented on cattle and hippo dung do not include any estimate of uncertainty. The differences in mean particle size and, for example, P content, are quite large and less of an issue, but differences in nitrogen, protein content, and other micronutrients are not as obvious and one wonders whether these are indeed significantly different from one another. Statements on, e.g., lines 222-226 could be overstating these results – and this needs to be acknowledged in the manuscript text.

A lot of weight is placed on the differences in lag/temporal dynamics and maximum rates (K) of GPP in response to the dung treatments. Yet, a close read of the supplementary material suggests that these relatively strong patterns resulted only after the theta values were changed to a higher value (1.1085 vs. 1.045). Given that the value used was atypical, this is a bit worrisome for the reader. The authors justify this choice by saying that (a) more days could be modeled with new value (but it's only 15% fewer days out of hundreds), and (b) ER did not show expected temperature dependence with the 1.045 value. Yet, temperature is not clearly reported in the paper (when it is, it is hard to see in the figure and the units are not shown). Thus, it's not clear that we should even expect big swings in ER driven by temperature in this ecosystem. I don't think this is a fatal issue for the paper, but the GPP results are quite central to the story. The authors need to be more upfront about this choice and its potential implications in the main text – not the supplementary material. And, the authors need to consider or acknowledge whether the conclusions would change significantly if the 1.045 value was used.

A thorough check of the supplementary material is needed for clarity/mistakes– it is somewhat sloppy at the moment. There are many spelling/grammatical, omissions errors. I would also suggest avoiding the inclusion of output graphs from R with axis labels that are not interpretable to the reader. Some of the lines on the graphs are also very difficult to see (e.g., temperature and light intensity in Figure S5).

Figure S3 is the only place the symbols refer to the percentage of hippo dung. All the others show percentages of cattle dung. Suggest changing this to represent cattle dung.

Minor:

Line 23: change 'ecosystem' to 'ecosystems'

Line 45: change 'are' to 'can be'

Line 48: this is a very general paragraph and this statement makes it seem that all ecosystems have a 'dry season'.

Lines 117-120: not clear what 'interaction' you are talking about that would lead to non-linearity.

Line 142: change to: 'the loading of cattle and hippopotamus dung'

Line 143: change 'distribution' to 'distributions'

Line 223: delete the word 'remarkably'. It overstates the differences

Line 235-236: delete what is in parentheses.

Line 247: due to differential leaching rates. . but also uptake and retention in biomass?

Line 249: reporting nitrite dynamics seems odd given what a small proportion of the pool it represents – and it's reduced importance in biological activity.

Figure 2 legend, line 308: delete 'on'

Decision letter (RSPB-2019-3000.R0)

14-Feb-2020

Dear Dr Masese,

Your manuscript has now been peer reviewed and the reviews have been assessed by an Associate Editor. The reviewers' comments (not including confidential comments to the Editor) and the comments from the Associate Editor are included at the end of this email for your reference. As you will see, the reviewers and the Editors have raised some concerns with your manuscript and we would like to invite you to revise your manuscript to address them.

We do not allow multiple rounds of revision so we urge you to make every effort to fully address all of the comments at this stage. Please pay particular attention to the presentation and check the manuscript over thoroughly before resubmission. If deemed necessary by the Associate Editor, your manuscript will be sent back to one or more of the original reviewers for assessment. If the original reviewers are not available we may invite new reviewers. Please note that we cannot guarantee eventual acceptance of your manuscript at this stage.

Research ethics:

Use of animals and field studies:

Please submit a copy of your revised paper within three weeks. If we do not hear from you within this time your manuscript will be rejected. If you are unable to meet this deadline please let us know as soon as possible, as we may be able to grant a short extension.

Best wishes,
 Professor Loeske Kruuk
 mailto: proceedingsb@royalsociety.org

Associate Editor Board Member

Comments to Author:

Both reviewers and I all agree that there has been a great deal of progress on this manuscript. However, reviewer 2 raises some crucial issues, especially the lack of an estimate of uncertainty. There is also an issue about clarifying the lag and temporal dynamics. These are important issues that need to be addressed comprehensively. Reviewer 1 also has a number of small changes that will improve the manuscript.

Reviewer(s)' Comments to Author:

Referee: 1

Comments to the Author(s).

General comment:

I have now read the manuscript by Masese et al. that was resubmitted to Proceedings B. I found the manuscript well-written, straightforward and relevant for a general audience already in the first round, and this hasn't changed. The flow of ideas is again very nice, and the study combines many different perspectives to arrive to the conclusion that cattle subsidies to aquatic ecosystems are fundamentally different than hippo loadings. This has consequences for ecosystems that are increasingly being influenced by cattle grazing. Nearly all of the issues I highlighted on the first round of revision have been addressed. Some bits of the manuscript have been toned down a bit (e.g. the title), but I don't think these changes detract any relevance or interest. I will specify some areas of improvement below, but all of my comments are very specific. The manuscript is of high quality in general. I would recommend acceptance with minor revisions.

Specific comments:

Abstract

- There is quite a lot of emphasis to the new estimates of cattle vs. hippo loadings. While these estimates are interesting, in the sense that they set a context to assess the relevance of this manuscript's results, they are 'big numbers' (see other comments below). Therefore, I would recommend giving them less emphasis in the abstract.

Introduction

- Lines 74-77. You say that 'Ruminants such as cattle [...] have a relatively efficient digestive system compared to non-ruminants...', then you also say that 'non-ruminants, such as hippos, have longer mean retention times than ruminants...'. However, in the Supplementary 1(a), you say: 'both cattle and hippos have long mean gut retention times'. Which of these sentences is the correct one? And importantly, if hippo and cattle gut retention times are different, as I would imagine, did you model cattle loadings using the gut retention times of hippos, as I gather from reading Supplementary 1(a) and Subalusky et al 2015 Freshwater Biology paper?
 - Apart from the need to clarify the previous point, the introduction is great.

Methods

- Line 129. A reference to support this method to estimate protein would be useful here.
 - Lines 136-137. I would recommend you to make it very clear to the reader that in this paper you only modelled cattle loadings, because hippo loadings had already been calculated in Subalusky et al 2015. Otherwise you end up a bit lost trying to find out how you calculated hippo loadings.

E.g. ‘... and compared results with EXISTING estimateS of loading rates for hippos in the river EXTRACTED FROM REF. [12]’.

- It would be important to acknowledge somewhere that the estimates of the loading rates for cattle and hippos are quite broad or ‘crude’. E.g. hippo data comes from captive hippos, the authors assume the same movement patterns for cattle during wet and dry seasons, and between protected (with predators) and unprotected (with less predators, I guess) areas...

- Line 186. You don’t need to apologise for not using GAMMs, hehe, especially if you fulfil the requirements to apply a less complex and ‘more parsimonious’ method (linear mixed effects models). I would just say that you used LMEMs and that you fulfilled normality and homoscedasticity.

- Line 195. Introduce why you want to fit the sigmoid Gompertz. E.g. ‘To test the effect of cattle dung on ecosystem respiration, or production or...’. Otherwise the paragraph appears a bit out of the blue.

Results

- Line 222. Try to use the past for all results. Here, I would write: ‘Cattle dung haD remarkably...’

- Lines 235-236. Can you clarify these results? There’s an question mark that appears out of place. I think this paragraph is an interesting addition, although I believe these estimates are quite grosso modo.

- Lines 252-253. If these variables increased with dung %, the coefficients in table 1 for dung should be positive. I see in the plots (FigS4) that the lines do increase, so the negative sign (Table1) must be an error, I guess.

- Line 266 and Fig. 1c. I think there has been a misunderstanding here. According to what I see in Fig. 1a, the effect of cattle dung was to increase K and ALSO INCREASE the rate at which K is achieved. However, the text and Fig 1c don’t agree with this. I think the issue is that according to Supplementary Information 4, page 11, after equation (2), we can read: ‘P2 (m² min⁻¹ g⁻¹ O₂) is the inverse maximum photosynthesis rate’. Therefore, if the rate plotted in Fig.1c, is the inverse of the rate at which K is achieved, everything makes sense again. And the text should be amended as follows: ‘... GPP increased FASTER AND reached a higher maximum (Fig. 1ab)’.

- Line 267. Reference to Fig. 1c should be FIG.1D.

- Lines 266-269. I don’t think you need to justify this result, because I really can’t see differences in the lag.

- Figure 2. Why didn’t you plot lines on figure 2? (as done in figures S2 or S4).

- Line 288. Can you explain how you get to the conclusion that blocks were not important, based on Table S4. I don’t know if the values present in the line starting Block (EDF(F)) are the p-values, or not...

- Figure legend of Fig. 1. After the last word, ‘cattle’ add ‘dung.’.

- Figure legend on Fig. 2. Line 308, delete ‘on’.

Discussion

- Line 355. Here you say it correctly according to what can be seen in Fig 1a (see previous comment about the rate in plot Fig.1c, being the inverse of the rate, in fact).

- Line 356. I would write biofilm/periphyton biomass to make things easier to those that haven’t read the methods.

- Lines 361-367. You don’t really explain why you didn’t find increased respiration in 0% cattle dung (thus 100% hippo dung) treatments... as it would be expected and as hypothesised in the introduction.

- Lines 369-372. Without a reference to Supplementary Information 6, these conclusions are quite a leap from the results. I understand the space constraints, but it really means the average reader has to ‘have faith’ in what you say, unless they go and read the supplementary in depth.

- Nice discussion and conclusions!

Referee: 3

Comments to the Author(s).

This study presents results from a mesocosm experiment aimed at quantifying ecosystem responses to varying qualities of mammalian dung. The study design maintains a similar amount of dung in mesocosms, but varies the contribution of cattle vs. hippo. Each week, various ecosystem structural and functional components are measured and/or modeled. The authors find that cattle dung is a higher quality subsidy that positively influences nutrients and biomass, and cattle dung changes the character and temporal trajectory of DOC. The authors conclude that changes in LMH identity and numbers may thus have a large influence on river ecosystems, such as the Mara. The authors also construct a simple model to estimate total dung loading by cattle vs. hippos.

Overall, the study is well conceived and executed, and complements a number of other studies in the system – largely focused on hippos. The analysis appears comprehensive, but at times comes across as a bit overcomplicated or overparameterized – especially in the context of the temporal dynamics and magnitude of ecosystem metabolism. I do think this is an important contribution that should be of interest to many ecologists – and hope that my comments help to strengthen the presentation.

The data presented on cattle and hippo dung do not include any estimate of uncertainty. The differences in mean particle size and, for example, P content, are quite large and less of an issue, but differences in nitrogen, protein content, and other micronutrients are not as obvious and one wonders whether these are indeed significantly different from one another. Statements on, e.g., lines 222-226 could be overstating these results – and this needs to be acknowledged in the manuscript text.

A lot of weight is placed on the differences in lag/temporal dynamics and maximum rates (K) of GPP in response to the dung treatments. Yet, a close read of the supplementary material suggests that these relatively strong patterns resulted only after the theta values were changed to a higher value (1.1085 vs. 1.045). Given that the value used was atypical, this is a bit worrisome for the reader. The authors justify this choice by saying that (a) more days could be modeled with new value (but it's only 15% fewer days out of hundreds), and (b) ER did not show expected temperature dependence with the 1.045 value. Yet, temperature is not clearly reported in the paper (when it is, it is hard to see in the figure and the units are not shown). Thus, it's not clear that we should even expect big swings in ER driven by temperature in this ecosystem. I don't think this is a fatal issue for the paper, but the GPP results are quite central to the story. The authors need to be more upfront about this choice and its potential implications in the main text – not the supplementary material. And, the authors need to consider or acknowledge whether the conclusions would change significantly if the 1.045 value was used.

A thorough check of the supplementary material is needed for clarity/mistakes– it is somewhat sloppy at the moment. There are many spelling/grammatical, omissions errors. I would also suggest avoiding the inclusion of output graphs from R with axis labels that are not interpretable to the reader. Some of the lines on the graphs are also very difficult to see (e.g., temperature and light intensity in Figure S5).

Figure S3 is the only place the symbols refer to the percentage of hippo dung. All the others show percentages of cattle dung. Suggest changing this to represent cattle dung.

Minor:

Line 23: change 'ecosystem' to 'ecosystems'

Line45: change 'are' to 'can be'

Line48: this is a very general paragraph and this statement makes it seem that all ecosystems have a 'dry season'.

Lines 117-120: not clear what 'interaction' you are talking about that would lead to non-linearity.
 Line 142: change to: 'the loading of cattle and hippopotamus dung'
 Line 143: change 'distribution' to 'distributions'
 Line 223: delete the word 'remarkably'. .it overstates the differences
 Line 235-236: delete what is in parentheses.
 Line 247: due to differential leaching rates. . but also uptake and retention in biomass?
 Line 249: reporting nitrite dynamics seems odd given what a small proportion of the pool it represents – and it's reduced importance in biological activity.
 Figure 2 legend, line 308: delete 'on'

Author's Response to Decision Letter for (RSPB-2019-3000.R0)

See Appendix B.

Decision letter (RSPB-2019-3000.R1)

26-Mar-2020

Dear Dr Masese

I am pleased to inform you that your manuscript RSPB-2019-3000.R1 entitled "Hippopotamus are distinct from domestic livestock in their resource subsidies to and effects on aquatic ecosystems" has been accepted for publication in Proceedings B.

The Associate Editor has recommended publication, but has also suggested some minor revisions to your manuscript. Therefore, I invite you to respond to his comments and revise your manuscript. Because the schedule for publication is very tight, it is a condition of publication that you submit the revised version of your manuscript within 7 days. If you do not think you will be able to meet this date, especially given the current conditions relating to COVID019, please let us know.

1) A text file of the manuscript (doc, txt, rtf or tex), including the references, tables (including captions) and figure captions. Please remove any tracked changes from the text before submission. PDF files are not an accepted format for the "Main Document".

2) A separate electronic file of each figure (tiff, EPS or print-quality PDF preferred). The format should be produced directly from original creation package, or original software format. PowerPoint files are not accepted.

3) Electronic supplementary material: this should be contained in a separate file and where possible, all ESM should be combined into a single file. All supplementary materials accompanying an accepted article will be treated as in their final form. They will be published alongside the paper on the journal website and posted on the online figshare repository. Files on figshare will be made available approximately one week before the accompanying article so that the supplementary material can be attributed a unique DOI.

Finally, all the best for dealing with the difficulties of the current global situation; I hope you are all well and stay safe.

Yours sincerely,

Professor Loeske Kruuk
Editor, Proceedings B
mailto:proceedingsb@royalsociety.org

Associate Editor:

Board Member

Comments to Author:

Thank you for your hard work on this manuscript and the response to reviewers. The manuscript has greatly improved based on both your work and the careful attention by the reviewers. I was a bit disappointed by the authors deciding that the uncertainty estimation was too difficult, but I recognize there are limitations to the data sources that may preclude that at this stage.

The one point I noticed on re-reading is that the towards the end of the discussion, the conceptual picture of this area of the literature does not come precisely into focus, as well as it does in the introduction.

For example the line 416: "More studies are needed to assess the spatial and temporal dynamics of dung input into aquatic ecosystems by not only cattle, but other livestock such as goats and sheep, and assess in situ ecosystem responses to these inputs. " Yes, of course more empirical studies are always needed, but it would be good at this stage of the manuscript to more precisely define the goal of this area of the literature. What synthetic understanding do we hope emerges from this (and other similar studies) in 5 years time?

The anthropogenic change to the world's herbivores has been huge both in the amount and the species composition. Two recent papers may be relevant for this big picture framing:

Bar-On, Yinon M., Rob Phillips, and Ron Milo. "The biomass distribution on Earth." *Proceedings of the National Academy of Sciences* 115.25 (2018): 6506-6511.

Berendes, David M., et al. "Estimation of global recoverable human and animal faecal biomass." *Nature Sustainability* 1.11 (2018): 679-685.

Author's Response to Decision Letter for (RSPB-2019-3000.R1)

See Appendix C.

Decision letter (RSPB-2019-3000.R2)

31-Mar-2020

Dear Dr Masese

I am pleased to inform you that your manuscript entitled "Hippopotamus are distinct from

domestic livestock in their resource subsidies to and effects on aquatic ecosystems" has been accepted for publication in Proceedings B.

Open Access

Paper charges

Sincerely,

Appendix A

Wild mammalian herbivores are distinct from domestic livestock in their resource subsidies and ecosystem effects

Associate Editor, Comments to Author:

We have now received two reviews of this paper on this manuscript on the subsidies by cows and hippos to aquatic ecosystems. Both reviewers found a great deal of potential in this manuscript, and I agree. Unfortunately there very serious conceptual points that are not well dealt with in the current manuscript.

The title doesn't reflect the paper very well. First, it creates the impression that this will be a general review of all wild mammalian herbivores, which the paper is definitely not. Hippos are, of course, an example of mammalian herbivores, but the reader cannot and should not extrapolate from hippos to other mammalian herbivores based on the evidence in this paper. In theory, the authors could build this argument in the discussion, perhaps based on C:N:P ratios of other wild mammalian herbivores. In fact the evidence cited about the East African Buffalo works against the authors own argument. (In this section the authors also mis-use the term "foregut fermenters", which is a group includes ruminants and hippos, ie ruminants definitely ferment in their foreguts.)

Response: Title: We have replaced "wild mammalian herbivores" with "Hippopotamus" in the title.

C:N:P ratios: We have included a supplementary table (Table S3) reporting some C:N:P ratios of common large mammalian herbivores in African savanna, and we discuss the implications of these differences in stoichiometry among LMH in the discussion. We have replaced the example of buffalo with cattle.

Mis-use of foregut fermenters: We have replaced foregut fermenters with 'non-ruminants' and defined cattle as ruminants

The other major conceptual issue raised by reviewer 2 is about mass of added organic material. This is a very serious shortcoming, especially given recent evidence about the effects of hippo dung published by one of the co-authors (Dutton et al. 2018). It seems likely that per unit mass of forage hippos are producing more mass of dung than cattle and that sheer quantity of organic matter is having an effect on aquatic ecosystems, an effect that is not captured by the authors' experiments.

Response: We agree that per unit mass of forage hippos are producing more mass of dung than cattle (see Supplementary Information 1) and that the quantity of organic matter has an effect on aquatic ecosystems. However, the number of cattle is more than an order of magnitude greater than the number of hippos, yielding comparable levels input in some places. We now include a model to estimate loading rates by cattle in the region and compare them to hippos, and we show that cattle inputs can range from 6-57 % of inputs relative to hippos (Supplementary Information 1). Our experimental design captures this range, as well as the endpoints of all cattle or all hippo loading.

Together these points suggest that main conceptual result in this manuscript is about the effect of ruminants versus pseudo-ruminants per unit feces, which is interesting, but it's not globally applicable in the way that a general comparison of wild mammals and livestock would be. (This point is also made by reviewer 1.)

Response: Our experimental design was intentionally developed to 1) compare the effects of hippo versus cattle dung on a per unit faeces basis, as this is necessary to isolate the effects of dung quality from those of dung quantity, and 2) analyze the effect of replacing hippos with cattle, which has been

reported in other areas in East Africa (Ogutu et al., 2016; Veldhuis et al., 2019), and is likely representative of what is happening in other savanna landscapes undergoing change.

Certainly the ecosystem level effects of cattle and hippos are important and interesting, but there is a serious limitation to how much a per unit feces experiment tells us. My recommendation is two-fold: first, the title and some of the framing of the paper needs to change to reflect that it's not reasonable to extrapolate to all wild mammalian herbivores. Second, these results need to be embedded in an ecosystem model to actually be informative about the effect of these two species on ecosystems. This model does not have to be complicated, in fact it's better if it's not, but there is certainly reason to believe that the per unit feces results might be out-weighted by difference in the total amount of subsidy. I realize this is a difficult request, but otherwise the results of this experiment alone are not enough to understand the effect of these two species on the ecosystem.

Title: *We have changed the title and replaced 'wild mammalian herbivores with "hippopotamus". We have also made some text changes to indicate that the findings of the experiment are not to be extrapolated to all wild mammalian herbivores.*

Ecosystem model and the role of subsidy quantity vs quality: *We have now developed an ecosystem model to estimate loading rates of organic matter (dung) by cattle and hippopotamus in the Mara River (Supplementary Information 1). Our model show that cattle inputs can range from 6-57 % of total inputs by cattle and hippos, and in some areas without hippos or without cattle but with hippos, the range is from 0-100% cattle dung or hippo dung relative to each other. It is important to note that the number of livestock outside the MMNR is highly variable ranging from 100,000 to 250,000 depending on the amount of rainfall and season (Lamprey and Reid, 2003), and the total number of cattle in the MRB is quite high (Hoffman et al., 2011; Ogutu et al., 2016; Veldhuis et al., 2019). Moreover, by varying the ratio of cattle dung to hippo dung, hence different quantities of both hippo dung and cattle dung, our experimental design does test the effect of both dung quality and quantity.*

Reviewers' Comments to Author:

Referee: 1

GENERAL COMMENTS

I have now read the manuscript by Masese et al. The manuscript experimentally assesses how terrestrial subsidies to aquatic ecosystems vary in quality along a gradient from 100% hippo dung to 100% cattle dung, as well as their effects on nutrients, organic matter, and ecosystem metabolism. They found strong differences of dung treatment on dissolved nutrients, organic matter, biomass of biofilms, chlorophyll, ecosystem metabolism and DOM composition. Gross primary production was increased by cattle dung, likely due to its higher quality (higher C:N ratio), and smaller particle size; in contrast, hippo dung tended to sink, and decreased primary production. Curiously enough, these effects on primary production were not mirrored by changes in ecosystem respiration. These results are relevant in the current context of increasing densities of cattle in many regions where wild mammalian herbivores used to be the main agent subsidising aquatic ecosystems. It is also relevant to further prove that not all mammalian herbivores influence the same ecosystem processes in the context of rewilding.

Response: *We appreciate the positive view of our study findings and their significance in understanding the potential role of increasing livestock numbers on aquatic ecosystems. On why the findings of increasing gross primary production with increasing amounts of cattle dung were not mirrored by changes in ecosystem respiration, we have a couple of potential explanations, which we have also captured in the revised ms- Ln 366-362.*

I think the authors have done a great job in summarising a huge amount of results of different nature (from dissolved nutrients, to DOM quality, to ecosystem metabolism), and analysed with very different techniques, which has produced a manuscript that is straightforward, easy-to-read and relevant for a general audience. I do have some concerns, of course, that I will specify below, but none of them is of a general nature – they are very specific issues that should not be difficult to address in a revised version. I therefore recommend the acceptance of this manuscript with minor revision.

Response: We appreciate your comment and would like to indicate that all the specific and general comments you have provided have been considered individually, and have considerably improved the revised ms.

SPECIFIC COMMENTS

Response: All specific comments have been implemented as suggested, except in cases that we have provided a responses as to why that is not the case.

Materials and methods.

Lines 128-131. It is a bit difficult to imagine the experimental design without Figure S1. I would therefore suggest moving it from the supplementary to the main manuscript (if space allows).

Response: A major limitation in preparing this ms is the number of words and the maximum number of pages (10) that a published article will not exceed. For this reason, it has been a delicate balancing act to decide which material goes to supplementary and which to retain in the main ms. Because we are operating at maximum number of words currently, it is not possible to add any additional information in the ms without replacing it with something.

Line 192. I think this is the first time 'DO' appears. It should be defined (I guess it's Dissolved Oxygen?).

Response: Done.

Line 197 and Supplementary Information 2. According to the authors, the temperature dependency of ecosystem respiration (θ , θ) is a 'tricky' parameter in metabolic models. The authors state that it is quite an important parameter, and that there has been a lot of discussion around it. In this sense, it would therefore be useful to conduct a sensitivity analysis of this parameter maybe? Or at least to see the effects of tuning it on the shapes of the plots in Figure 2. E.g. how do the plots in fig 2 look like if the authors use the 'commonly used value of 1.045 instead of 1.1085?

Response: We have conducted a sensitivity analysis of theta by comparing model outputs using our modeled higher value of 1.1085 and the commonly used value of 1.045 – see details in Supplementary Information 3 (previously Supplementary Information 2). We have also compared plots in Figure 2 (Figures S3 and S4). The results show that while the patterns in GPP, GPP:ER and NEP are increasing with increasing amounts of cattle dung in both models (using either 1.1085 or 1.045), using a lower value of theta reduced the number of days that could successfully be modeled (only 54.9%) compared with 70.5% of days that were successfully modeled using a higher theta value (1.1085). The higher number of days that were successfully modeled using our chosen theta value of 1.1085 improved our identification of relationships and differences among dung treatments in this study.

How they ended up with this final value of 1.1085?

Response: After obtaining poor model outputs using the commonly used theta value of 1.045, we decided to model theta for our study. To arrive at this theta value (1.1085), we selected initial 50 days from different dung treatments and periods from among the 44 days experimental period (6 weeks) and modeled theta along with P1, P2 and ER (4-parameter model) and used a fixed reaeration coefficient (k)

that we measured in our mesocosms. An average theta value was then obtained from successfully modeled days (see below), which we then fixed for a 3-parameter model (P1, P2 and ER modeled and k and theta fixed) used to model all days of the experiment.

How did you assess the goodness of fit of the models? In other words, what does it mean the sentence 'our models were better constrained [with higher theta]'.

Response: A number of checks were done to pick the number of days that were successfully modeled (constrained model outputs) and whose results were used for subsequent analyses. First, we used nlm in the metabolism FIT function to minimize the negative log-likelihood between measured and modeled DO values. Low values (< -100) of sum of squared residuals for each model run were considered indicative of a successful and constrained fit. Secondly, model fits (graphs) were inspected to confirm that the modeled DO values perfectly or closely matched measured (observed) DO values (Figure S2). Finally, the modeled outputs for GPP and ER were inspected to make sure that they made sense. For instance, cases where GPP values were negative or ER values were zero or positive were discarded, including cases where outputs were too large or too low compared to values of previous successfully modeled days.

Line 199. It appears to me that you used Linear Mixed Effects models instead of GLMMs. I say this because you said you used the function lme() (line 200) and because you say you transformed some variables to meet normality assumptions (lines 205-6).

Response: This has been corrected.

Line 211. Can you please explain a little bit to the non-specialist reader what each of these parameters mean?

Response: Done – Ln 198-203.

Line 212. Why did you pool data from 3 streams? Is that the reason why you don't get error bands in Fig 1a,e? But then why are there confidence intervals and error bars for the rest of the panels in Fig 1? Because these are the error bands coming from regression? This bit needs a bit of clarification, since you state you used both the data pooled and the data for each stream, but I'm not sure how this was done, in the end (I would say pooled).

Response: To fit the Gompertz model for each dung treatment, we used individual replicates (what we are calling pooled data). This produced graphs 1a and 1e. However, we also ran Gompertz model for each stream (3 replicates = 3 streams per dung treatment) and the parameters (K, rate and lag) obtained from each stream were used to calculate means for each dung treatment, and hence the error bars in 1b-d and 1f-h.

Results.

Line 234. 'Cow' should be replaced by 'cattle'.

Response: Done

Line 245. 'Figure S2a,b' should be 'Figure S2a,c', I think.

Response: Done

Line 246. 'Figure S2c,d' should be 'Figure S2b,d', I think.

Response: Done

Line 275-276. Was this because you pooled data from the 3 replicate streams?

Response: Yes, we used all the three replicates from each block in the plots

Line 283-284. ‘... further indicating that the positive effect of cattle dung on GPP increased with time.’ Where can we see these results? Is it in Supplementary information 4, Table S2?

Response: *Yes- reference has been made to the table.*

Table 1. These appear to be the results from doing summary(model) in R. These tests, check the significance of the parameters (whether their estimate cross the 0 line). To check the main effects, you should also conduct the test of main effects, using anova() or Anova() in R. Can you report these results?

Response: *We have added rows reporting the results of the anova*

Lines 290-296 need a bit of editing. Currently I find them difficult to follow.

Response: *The statement has been revised*

Line 292. I cannot see the non-linear trend (line 290) in Figure 2.

Response: *The statement has been revised.*

Line 293. It should be Supporting Information 4, not 5, I think.

Response: *This has been revised accordingly to reflect addition of Supplementary Information 1 on the ecosystem model of cattle vs hippopotamus loading of organic matter (dung) into the Mara River.*

Figure 1. The labels (a-h) for each panel are lacking.

Response: *Fixed.*

Line 316. Supporting Information 5, not 6, I think.

Response: *This has been revised accordingly.*

Line 324-326. Where can I see the results described here?

Response: *These results are from figure 3a,b. Reference has been made in the sentence to this figure*

Lines 326-328. Where can I see these results?

Response: *These results are from figure 3a,b. Reference has been made in the sentence to this figure*

Lines 330-331. Where can I see these results?

Response: *These results are from figure 3a,b. Reference has been made in the sentence to this figure*

Line 338. After ‘...cattle were slightly lower again’ should go (Figure 3c), I think.

You should also describe Figure 3b. Currently not described in the results section.

Response: *Fixed. Figure 3b has been cited in text along with Figure 3a which are interpreted together.*

Discussion.

Line 362. Replace ‘accumulates’ for ‘accumulate’.

Response: *Fixed.*

Line 367. ‘Replace’ increasing by ‘increased’.

Response: *Fixed.*

Line 369. Figure 1b, should be Figure 1c.

Response: *Fixed.*

Line 371. Figure 1b, should be Figure 1c.

Response: *Fixed.*

Lines 374-384. This paragraph is great, but it’s very difficult for the general readers to jump from the results section to these conclusions if you don’t give them a bit more context (either in the results, or here, or supplementary).

Response: *Because of space limit is the ms because of number of words, this has been done in the supplementary section*

Line 378. Figure 3 should be referenced here, instead of Figure 2, I think.

Response: *Fixed.*

Lines 428. ‘... reduced benthic primary production...’ compared to the treatments with high % of cattle

dung treatments. What about compared to water without ‘subsidies’? You didn’t have this treatment in this study, but it would be interesting to discuss somewhere in the discussion what is the overall effect of large herbivorous mammal subsidies in streams. And then talk about how domestic vs. wild large herbivores influence differently ecosystem processes, which is the focus of this study.

Response: We appreciate the addition suggested, which we have added to the sentence. However, we did not have a ‘control’ in this experiment, that is ‘streams without subsidies’, because the focus was on comparing the two herbivores because of their overlapping distribution in the Mara River basin and other river systems (Field, 1970; Mosepele et al., 2009; De longh et al., 2011; Kanga et al., 2013; Hulot et al., 2019). Furthermore, other work by several authors on this paper already compared the effect of a range of hippo inputs to a control (Subalusky et al. 2018)(Dutton et al. 2018 also has hippo addition effects), and we are building upon that work here. We have added some text to the introduction to make this clear. We have also added more information on the discussion on the overall effect of large herbivorous mammal subsidies in streams (Lines 398-409).

Referee: 2

GENERAL COMMENTS

Reviewer comments for Masese et al. submitted to Proceedings B, “Wild mammalian herbivores are distinct from domestic livestock in their resource subsidies to and effects on aquatic ecosystems”
This manuscript describes results of a study comparing the effects of hippo versus cattle dung on recipient aquatic ecosystems. The study used experimental mesocosms amended with a wide range of combinations of hippo and cattle dung, establishing a gradient to investigate interactive and singular effects of each on water chemistry, periphyton, ecosystem metabolism, and dissolved organic matter. The study shows that nutrient-rich cattle dung drive greater autotrophy within recipient systems, with implications for nutrient dynamics, periphyton growth, gross primary production, and the chemistry of dissolved organic matter. The intensive characterization, extended duration, and wide treatment gradient are real strengths of the study – as are the clear implications for aquatic ecosystems under global change. I do have recommendations for revision of the manuscript, below.

Major comments:

The experimental design is a nice test of the effects of cattle vs. hippo dung on aquatic ecosystems, but I would have liked to see an experimental control treatment (no dung added) or a gradient of absolute quantity added, instead of different quantities of each in combination. This would provide a picture of a broader question of how dung affects recipient systems, and how much the effects of cattle dung specifically compare to effects of simply adding any dung. Is there a reason you did not include a control treatment, and would you address this in your discussion?

Response: While we recognize the role of a control (no dung addition) in these experiments, prior research by members of our team has already addressed this question in a similar experimental design (Subalusky et al. 2018). In this paper, we are testing the ecosystem responses to displacement of wildlife by livestock in savanna systems, as this is the common land use change in these systems.

Based on the title and nature of this study, I was expecting to see more data quantifying the inputs of material (at least dung) from cattle versus hippos measured in the field. This seems like a major remaining knowledge gap, because it is not clear whether the experimental conditions in your study mimic the actual mode and quantity of inputs to aquatic ecosystems. The paragraph of lines 390-403 addresses these concerns well, but the authors should perhaps be more specific about what questions remain a priority for future studies, either resulting from their own findings...or to put the results in a more realistic context.

Response: We have developed a model for estimating the loading rates of cattle dung by cattle into streams and river in comparison to the loading rates of hippopotamus. Our model shows that livestock

inputs can range from 6-57% of inputs relative to hippos, thus suggesting our experimental design is reflective of this system. We have also added future research needs that need attention, i.e., "assessing the spatial and temporal dynamics of dung input into aquatic ecosystems by not only cattle, but other livestock such as goats and sheep, and assess ecosystem responses to these inputs".

Line comments:

Line 53 – Please break up this sentence by adding commas such as after grasslands, forages, and ecosystems.

Response: Fixed.

Lines 79-81 – Nice detail showing the contrasting C:N:P of dung from different large herbivores, but please qualify this sentence because the writing makes it seem these ratios are fixed, but dung C:N:P surely vary over time and across individuals from each species. For example, consider listing mean +/- SE or simply mention these data were measured from a subset of individuals.

Response: Fixed.

Line 200 – NO₃ has only 1 minus in its superscript

Response: Fixed.

Line 214 – I am not very familiar with the term "lag" for describing temporal changes in metabolism. Can you explain here what this term describes – for example, what do high versus low values of "lag" indicate?

Response: Lag is a dimensionless parameter for location of the sigmoid curve along the time axis, i.e., it shifts graph to the left or right and is related to the time taken to reach the upper asymptote (maximum GPP or ER), with high values indicating fast progression towards maximum GPP or ER. We have added this information in the ms.

Lines 241-246 – These temporal changes in nutrient concentrations are very interesting, but are difficult to interpret in the supplemental figures because the legend is time. Please consider making time the X-axis in these figures, as you have done in Figure 1a, to show the temporal dynamics across treatments.

Response: We have included an additional figure (Figure S6) with time on the X-axis

Line 366 – I agree that the effects of cattle dung may be more widespread, but the high capacity for transport suggests that these effects will also be more diffuse (not as high of magnitude or persistent at the site of subsidy, for example).

Response: The sentence has been revised accordingly

Lines 367-373 – This paragraph is short and feels under-developed. I would be interested to see more discussion of what the contrasting GPP vs. ER responses to cattle dung might mean. Do cattle dung more strongly stimulate algae than they stimulate heterotrophic components (bacteria/fungi) of the periphyton, for example? Perhaps cattle dung not only stimulate algal growth...but also increase algal health, such that there is more GPP per unit periphyton chlorophyll or ER, and the experiment did not run long enough for algal mortality to ensue and stimulate respiration?

Response: We have added the following explanation for the discrepancy in GPP and ER responses to dung treatment Ln 355-366: Cattle dung also increased AFDM in the water column, which is a cumulative measure of both microbial and algal production. However, there were no dung treatment effects on the fitted sigmoid parameters of ER. Hippo dung, which was comprised of larger particles, tended to sink to

the bottom of the streams and reduce benthic production, suggesting it may do the same in aquatic systems, especially during low flows [13, 15, 16]. In contrast, cattle dung tended to remain suspended or dissolve, and in aquatic systems it may become dispersed by river discharge into a larger area, creating potentially more widespread and diffuse effects. Because light is one of the key determinants controlling production and composition of periphyton or algae in aquatic ecosystems [57], it is likely that cattle dung more strongly stimulated the autotrophic component (algae) of periphyton while hippos stimulated the heterotrophic component (bacteria/fungi), which led to higher GPP per unit biomass of periphyton among cattle dung treatments.

Lines 385-389 – This paragraph also seems rather short. Some of this material could perhaps be combined with the previous 2 paragraphs. I find it quite interesting that potential DOM sources shift depending on the dung type, not only at the beginning of the study but also long-term (later in the study) by indirectly affecting autotrophy.

Response: *The paragraph has been combined with the previous one.*

Line 417 – unclear what “their” refers to. Cattle dung?

Response: *Sentence has been revised for clarity*

Lines 431-433 – This final sentence seemed out-of-place for the paper, because rewilding is not mentioned elsewhere in the paper, and there is not a clear connection to your specific study findings. For example, are cattle being used in rewilding efforts...or are hippos receiving more protections toward rewilding? Consider elaborating on these specifics in the introduction, or instead, focusing on broad contrasts between wildlife and livestock as mediators of subsidies from terrestrial to aquatic systems, e.g. returning to ecological implications of points raised in lines 60-73. Based on your experimental findings, how might replacement of wild herbivores by domesticated livestock change aquatic ecosystems...or are the 2 groups functionally quite similar?

Response: *We have removed the aspect on rewilding and instead focused on the broad contrasts between wildlife and livestock as mediators of subsidies from terrestrial to aquatic systems.*

Figure 1 – Panel letters are missing from the figure proper. In (b), (c), (f), and (g) – K and Rate have units of g O₂ per day? Are values per unit area (meter squared), as in Figure 2?

Figure 3 – I really like the arrows in panel (a). Consider adding a note in the figure title that the arrows designate the time series from early to late in experiment for each treatment. My interpretation is that the response variable “total DOM dynamics” in (c) describes the approximate ‘length’ of the temporal arrows in panel (a) – can a note to this effect also be added?

Response: *This has been done.*

References

- Acuna, V., Wolf, A., Uehlinger, U. & Tockner, K. 2008 Temperature dependence of stream benthic respiration in an Alpine river network under global warming. *Freshwater Biol* **53**, 2076-2088. (doi:10.1111/j.1365-2427.2008.02028.x).
- Demars, B.O., Thompson, J. & Manson, J.R. 2015 Stream metabolism and the open diel oxygen method: Principles, practice, and perspectives. *Limnology and Oceanography: Methods* **13**, 356-374.
- Guenet, B., Danger, M., Abbadie, L. & Lacroix, G. 2010 Priming effect: bridging the gap between terrestrial and aquatic ecology. *Ecology* **91**, 2850-2861.

- Bianchi, T.S. 2011 The role of terrestrially derived organic carbon in the coastal ocean: A changing paradigm and the priming effect. *Proceedings of the National Academy of Sciences* **108**, 19473-19481. (doi:10.1073/pnas.1017982108).
- Dutton, C. L., Subalusky, A. L., Hamilton, S. K., Rosi, E. J., & Post, D. M. (2018). Organic matter loading by hippopotami causes subsidy overload resulting in downstream hypoxia and fish kills. *Nature communications*, 9(1), 1951.
- Dawson, J., Pillay, D., Roberts, P.J. & Perissinotto, R. 2016 Declines in benthic macroinvertebrate community metrics and microphytobenthic biomass in an estuarine lake following enrichment by hippo dung. *Scientific reports* **6**, 37359. (doi:10.1038/srep37359)
- Subalusky, A.L., Dutton, C.L., Njoroge, L., Rosi, E.J. & Post, D.M. 2018 Organic matter and nutrient inputs from large wildlife influence ecosystem function in the Mara River, Africa. *Ecology* **99**, 2558-2574. (doi:10.1002/ecy.2509)
- Kanga, E. M., Ogutu, J. O., Piepho, H. P., & Oloff, H. (2013). Hippopotamus and livestock grazing: influences on riparian vegetation and facilitation of other herbivores in the Mara Region of Kenya. *Landscape and Ecological Engineering*, 9(1), 47-58.
- De longh, H. H., De Jong, C. B., Van Goethem, J., Klop, E., Brunsting, A. M. H., Loth, P. E., & Prins, H. H. T. (2011). Resource partitioning among African savanna herbivores in North Cameroon: the importance of diet composition, food quality and body mass. *Journal of Tropical Ecology*, 27(5), 503-513.
- Field, C. R. (1970). A study of the feeding habits of the hippopotamus (*Hippopotamus Amphibius* Linn) in the Queen Elizabeth National Park, Uganda, with some management implications. *African Zoology*, 5(1).
- Hulot, F. D., Prijac, A., Lefebvre, J. P., Msiteli-Shumba, S., & Kativu, S. (2019). A first assessment of megaherbivore subsidies in artificial waterholes in Hwange National Park, Zimbabwe. *Hydrobiologia*, 837(1), 161-175.
- Mosepele, K., Moyle, P. B., Merron, G. S., Purkey, D. R., & Mosepele, B. (2009). Fish, floods, and ecosystem engineers: aquatic conservation in the Okavango Delta, Botswana. *Bioscience*, 59(1), 53-64
- Ogutu, J.O., Piepho, H.-P., Said, M.Y., Ojwang, G.O., Njino, L.W., Kifugo, S.C. & Wargute, P.W. 2016 Extreme Wildlife Declines and Concurrent Increase in Livestock Numbers in Kenya: What Are the Causes? *Plos One* **11**, e0163249. (doi:10.1371/journal.pone.0163249).
- Veldhuis, M.P., Ritchie, M.E., Ogutu, J.O., Morrison, T.A., Beale, C.M., Estes, A.B., Mwakilema, W., Ojwang, G.O., Parr, C.L. & Probert, J. 2019 Cross-boundary human impacts compromise the Serengeti-Mara ecosystem. *Science* **363**, 1424-1428.

Appendix B

Wild mammalian herbivores are distinct from domestic livestock in their resource subsidies and ecosystem effects

14-Feb-2020

Dear Dr Masese,

Your manuscript has now been peer reviewed and the reviews have been assessed by an Associate Editor. The reviewers' comments (not including confidential comments to the Editor) and the comments from the Associate Editor are included at the end of this email for your reference. As you will see, the reviewers and the Editors have raised some concerns with your manuscript and we would like to invite you to revise your manuscript to address them.

We do not allow multiple rounds of revision so we urge you to make every effort to fully address all of the comments at this stage. Please pay particular attention to the presentation and check the manuscript over thoroughly before resubmission. If deemed necessary by the Associate Editor, your manuscript will be sent back to one or more of the original reviewers for assessment. If the original reviewers are not available we may invite new reviewers. Please note that we cannot guarantee eventual acceptance of your manuscript at this stage.

Associate Editor Board Member

Comments to Author:

Both reviewers and I all agree that there has been a great deal of progress on this manuscript. However, reviewer 2 raises some crucial issues, especially the lack of an estimate of uncertainty. There is also an issue about clarifying the lag and temporal dynamics. These are important issues that need to be addressed comprehensively. Reviewer 1 also has a number of small changes that will improve the manuscript.

Reviewer(s)' Comments to Author:

Referee: 1

Comments to the Author(s).

General comment:

I have now read the manuscript by Masese et al. that was resubmitted to Proceedings B. I found the manuscript well-written, straightforward and relevant for a general audience already in the first round, and this hasn't changed. The flow of ideas is again very nice, and the study combines many different perspectives to arrive to the conclusion that cattle subsidies to aquatic ecosystems are fundamentally different than hippo loadings. This has consequences for ecosystems that are increasingly being influenced by cattle grazing. Nearly all of the issues I highlighted on the first round of revision have been addressed. Some bits of the manuscript have been toned down a bit (e.g. the title), but I don't think these changes detract any relevance or interest. I will specify some areas of improvement below, but all of my comments are very specific. The manuscript is of high quality in general. I would recommend acceptance with minor revisions.

Specific comments:

Abstract

- There is quite a lot of emphasis to the new estimates of cattle vs. hippo loadings. While these estimates are interesting, in the sense that they set a context to assess the relevance of this manuscript's results, they are 'big numbers' (see other comments below). Therefore, I would recommend giving them less emphasis in the abstract.

Response: We have removed reference to and emphasis on the numbers and generally reported on the differences in loading rates between cattle and hippos as a results of differences in physiology, size and behaviour. The edited text reads (Ln 28-32): "Our loading model shows that per capita dung input by cattle is lower than for hippos, but total dung inputs by cattle comprise a significant portion of loading from large herbivores due to the large numbers of cattle on the landscape".

Introduction

- Lines 74-77. You say that 'Ruminants such as cattle [...] have a relatively efficient digestive system compared to non-ruminants...', then you also say that 'non-ruminants, such as hippos, have longer mean retention times than ruminants...'. However, in the Supplementary 1(a), you say: 'both cattle and hippos have long mean gut retention times'. Which of these sentences is the correct one? And importantly, if hippo and cattle gut retention times are different, as I would imagine, did you model cattle loadings

using the gut retention times of hippos, as I gather from reading Supplementary 1(a) and Subalusky et al 2015 Freshwater Biology paper?

Response: *Cattle and hippos have different retention times, but the retention times for both are long – that is more than a day (24 h). We have clarified this in the Supplementary Information (.).*

Loading models: *We assumed that cattle excretion and egestion occurred at a constant rate throughout the day, due to the very long gut passage time. Thus, their loading estimates were based on time budgets. This was a similar approach as was following in the Subalusky et al. 2015 Freshwater Biology paper, in which the actual gut passage time was not incorporated into the loading estimates directly.*

- Apart from the need to clarify the previous point, the introduction is great.

Response: *This appreciated!*

Methods

- Line 129. A reference to support this method to estimate protein would be useful here.

Response: *A reference has been given*

- Lines 136-137. I would recommend you to make it very clear to the reader that in this paper you only modelled cattle loadings, because hippo loadings had already been calculated in Subalusky et al 2015. Otherwise you end up a bit lost trying to find out how you calculated hippo loadings. E.g. ‘... and compared results with EXISTING estimateS of loading rates for hippos in the river EXTRACTED FROM REF. [12]’.

Response: *This has been done.*

- It would be important to acknowledge somewhere that the estimates of the loading rates for cattle and hippos are quite broad or ‘crude’. E.g. hippo data comes from captive hippos, the authors assume the same movement patterns for cattle during wet and dry seasons, and between protected (with predators) and unprotected (with less predators, I guess) areas...

Response: *We agree there are a number of sources of potential error in the loading models. However, we believe they accurately reflect the conditions in the system. We have included an acknowledgement of this potential error in the Discussion in Ln 414-415:*

“Our estimates of both hippo and livestock loading have some uncertainty around them that could be improved with more detailed knowledge of animal time budgets and population sizes.”.

- Line 186. You don’t need to apologise for not using GAMMs, hehe, especially if you fulfil the requirements to apply a less complex and ‘more parsimonious’ method (linear mixed effects models). I would just say that you used LMEMs and that you fulfilled normality and homoscedasticity.

Response: *We have made changes as suggested.*

- Line 195. Introduce why you want to fit the sigmoid Gompertz. E.g. ‘To test the effect of cattle dung on ecosystem respiration, or production or...’. Otherwise the paragraph appears a bit out of the blue.

Response: *We appreciate the suggestion. We have made changes as suggested.*

Results

- Line 222. Try to use the past for all results. Here, I would write: ‘Cattle dung haD remarkably...’

Response: Done.

- Lines 235-236. Can you clarify these results? There's a question mark that appears out of place. I think this paragraph is an interesting addition, although I believe these estimates are quite *grosso modo*.

Response: *We regret the error with the question mark and have rectified. These numbers show great variation because of the nature of cattle movements and their distributions on the landscape. As noted elsewhere in the paper (Ln 100-104), cattle are not supposed to be grazed in the protected MMNR, but illegal grazing of a small number of cattle occurs- hence the low numbers on OM input inside the reserve. The number of hippos inside and outside the reserve does not change much, but further away, the hippo number diminish, hence the high percentage of livestock inputs compared to that by hippos. The input and changes have been made in Ln 241-247*

- Lines 252-253. If these variables increased with dung %, the coefficients in table 1 for dung should be positive. I see in the plots (FigS4) that the lines do increase, so the negative sign (Table1) must be an error, I guess.

Response: *We agree about the positive coefficients and have made a correction.*

- Line 266 and Fig. 1c. I think there has been a misunderstanding here. According to what I see in Fig. 1a, the effect of cattle dung was to increase K and ALSO INCREASE the rate at which K is achieved. However, the text and Fig 1c don't agree with this. I think the issue is that according to Supplementary Information 4, page 11, after equation (2), we can read: 'P2 (m² min⁻¹ g⁻¹ O₂) is the inverse maximum photosynthesis rate'. Therefore, if the rate plotted in Fig.1c, is the inverse of the rate at which K is achieved, everything makes sense again. And the text should be amended as follows: '... GPP increased FASTER AND reached a higher maximum (Fig. 1ab)'.

Response: *We note the concerns of the reviewer here , but the results in figure 1 and the modelling in Supplementary Information 4 are different. The Gompertz sigmoid parameters (K, rate and lag) are based on the GPP and ER estimates, and are not results from the modelling in SI 4. We therefore retain the reporting as it is.*

- Line 267. Reference to Fig. 1c should be FIG.1D.

Response: *Agreed and fixed.*

- Lines 266-269. I don't think you need to justify this result, because I really can't see differences in the lag.

Response: *The differences are statistically significant, so we retain the justification.*

- Figure 2. Why didn't you plot lines on figure 2? (as done in figures S2 or S4).

Response: *We tested for linearity and the responses in metabolism (GPP and ER) were not linear, so we performed GAMMs instead of LMEMs, and hence did not fit the lines.*

- Line 288. Can you explain how you get to the conclusion that blocks were not important, based on Table S4. I don't know if the values present in the line starting Block (EDF(F)) are the p-values, or not...

Response: *The relevant statistics for Block are EDF = estimated degrees of freedom, and the F value in brackets. The p-values are not indicated in the table, but where there are significant differences, asterisks (*) have been used. For Block, the effect is not statistically significant.*

- Figure legend of Fig. 1. After the last word, 'cattle' add 'dung'.

Response: *done*

- Figure legend on Fig. 2. Line 308, delete 'on'.

Response: *done*

Discussion

- Line 355. Here you say it correctly according to what can be seen in Fig 1a (see previous comment about the rate in plot Fig.1c, being the inverse of the rate, in fact).

Response: *We have explained above that the two are different*

- Line 356. I would write biofilm/periphyton biomass to make things easier to those that haven't read the methods.

Response: *Change has been made*

- Lines 361-367. You don't really explain why you didn't find increased respiration in 0% cattle dung (thus 100% hippo dung) treatments... as it would be expected and as hypothesised in the introduction.

Response: *It is likely that increased ER from heterotrophs in the hippo treatment was offset by the increased autotrophic respiration in the cattle treatments.*

- Lines 369-372. Without a reference to Supplementary Information 6, these conclusions are quite a leap from the results. I understand the space constraints, but it really means the average reader has to 'have faith' in what you say, unless they go and read the supplementary in depth.

Response: *The limitation imposed by the length of the paper is a major challenge, but we have tried our best to communicate as much as we can within the limit.*

- Nice discussion and conclusions!

Response: *Appreciated*

Referee: 3

Comments to the Author(s).

This study presents results from a mesocosm experiment aimed at quantifying ecosystem responses to varying qualities of mammalian dung. The study design maintains a similar amount of dung in mesocosms, but varies the contribution of cattle vs. hippo. Each week, various ecosystem structural and functional components are measured and/or modeled. The authors find that cattle dung is a higher quality subsidy that positively influences nutrients and biomass, and cattle dung changes the character and temporal trajectory of DOC. The authors conclude that changes in LMH identity and numbers may thus have a large influence on river ecosystems, such as the Mara. The authors also construct a simple model to estimate total dung loading by cattle vs. hippos.

Overall, the study is well conceived and executed, and complements a number of other studies in the system – largely focused on hippos. The analysis appears comprehensive, but at times comes across as a bit overcomplicated or overparameterized – especially in the context of the temporal dynamics and magnitude of ecosystem metabolism. I do think this is an important contribution that should be of interest to many ecologists – and hope that my comments help to strengthen the presentation.

The data presented on cattle and hippo dung do not include any estimate of uncertainty.

Response: *We have provided these estimates of mean loading rates in this manuscript solely to provide context for the experiment, and providing additional details to allow extrapolation of loading error is outside the scope of this paper. However, we have added some additional changes in recognition of the limitations of these estimates:*

- In the abstract we have removed the reference to and emphasis on the numbers and generally reported on the differences in loading rates between cattle and hippos, which we attribute to differences in their physiology, size and behaviour. The edited text reads (Ln 28-30): “Our loading model shows that per capita dung input by cattle is lower than for hippos, but total dung inputs by cattle comprise a significant portion of loading from large herbivores due to the large numbers of cattle on the landscape”.

-In the discussion (Ln 418-420), the uncertainty has been addressed by observing that “ Our estimates of both hippo and livestock loading have some uncertainty around them that could be improved with more detailed knowledge of animal time budgets and population sizes”.

The differences in mean particle size and, for example, P content, are quite large and less of an issue, but differences in nitrogen, protein content, and other micronutrients are not as obvious and one wonders whether these are indeed significantly different from one another. Statements on, e.g., lines 222-226 could be overstating these results – and this needs to be acknowledged in the manuscript text.

Response: *We note that some specific values in macro- and micro-nutrient composition between cattle dung and hippo dung, as presented in Table S3, appear to be close. However, the differences we report are quite consistent with findings in other studies (e.g., Table S2), which reinforces our conclusion that overall the quality of hippo dung is lower than that of cattle dung. This is also reinforced by the findings of the study. We have removed the word “remarkably” from the first sentence so as not to overestimate the differences (Ln 229).*

A lot of weight is placed on the differences in lag/temporal dynamics and maximum rates (K) of GPP in response to the dung treatments. Yet, a close read of the supplementary material suggests that these relatively strong patterns resulted only after the theta values were changed to a higher value (1.1085 vs. 1.045). Given that the value used was atypical, this is a bit worrisome for the reader. The authors justify this choice by saying that (a) more days could be modeled with new value (but it's only 15% fewer days out of hundreds), and (b) ER did not show expected temperature dependence with the 1.045 value. Yet, temperature is not clearly reported in the paper (when it is, it is hard to see in the figure and the units are not shown). Thus, it's not clear that we should even expect big swings in ER driven by temperature in this ecosystem. I don't think this is a fatal issue for the paper, but the GPP results are quite central to the story. The authors need to be more upfront about this choice and its potential implications in the main text – not the supplementary material. And, the authors need to consider or acknowledge whether the conclusions would change significantly if the 1.045 value was used.

Response: *We agree that using different values of theta resulted in different values for GPP and ER, and this was the essence of the sensitivity analysis we did to determine whether the same conclusion could be reached with the atypical theta value of 1.108. As we have reported in the Supplementary Information 4 on sensitivity analysis, and in Figures S6 and S7, the trends and trajectory of change in both GPP and ER are the same for both theta values. Thus, the conclusions reached in this study will not change even with the use of the common value of theta (1.045). We have explained the implications of using a different theta value in the results (Ln 277-281) and in the discussion – Ln 374-376.*

A thorough check of the supplementary material is needed for clarity/mistakes– it is somewhat sloppy at the moment. There are many spelling/grammatical, omissions errors. I would also suggest avoiding the inclusion of output graphs from R with axis labels that are not interpretable to the reader. Some of the lines on the graphs are also very difficult to see (e.g., temperature and light intensity in Figure S5).

Response: *We appreciate this observation and have efforts to improved the clarity of the text and quality of the figures, including axis labels.*

Figure S3 is the only place the symbols refer to the percentage of hippo dung. All the others show percentages of cattle dung. Suggest changing this to represent cattle dung.

Response: *Figure S2 and S3 present the same results on nutrients, whereby Figure S2 shows percentage of cattle dung as the rest of the graphs. Figure 3 has been presented to emphasize the effect of time on nutrient dynamics- i.e., the disappearance of SRP with time and the shift between nitrite and nitrate as reported in the paper in Ln 252-255.*

Minor:

Line 23: change 'ecosystem' to 'ecosystems'

Response: *Fixed*

Line45: change 'are' to 'can be'

Response: *Fixed*

Line48: this is a very general paragraph and this statement makes it seem that all ecosystems have a 'dry season'.

Response: *We have removed the 'dry season' part and added 'in rangelands and pastoralist areas'*

Lines 117-120: not clear what 'interaction' you are talking about that would lead to non-linearity.

Response: *Here, we were referring to mixture effects whereby cattle dung and hippo dung mixtures could support higher rates of both GPP and ER than a pure hippo or cattle dung treatment. Mixture effects and interactions have been obtained in litter decomposition experiments whereby, 'sharing' of resources between the component litters of a mixture facilitates the degradation of one litter by the presence of another (McTiernan, Ineson & Coward, 1997; Gartner, & Cardon2004), leading to higher decomposition rates in litter mixtures of different species compared to individual leaf species. However, this hypothesis is not supported in this study as higher fractions of cattle dung led to higher rates of GPP. Thus, we have not pursued this hypothesis elsewhere in the paper.*

Line 142: change to: 'the loading of cattle and hippopotamus dung'

Response: *Fixed*

Line 143: change 'distribution' to 'distributions'

Response: *Fixed*

Line 223: delete the word 'remarkably'. .it overstates the differences

Response: *Fixed*

Line 235-236: delete what is in parentheses.

Response: Fixed

Line 247: due to differential leaching rates. . but also uptake and retention in biomass?

Response: *The addition has been made*

Line 249: reporting nitrite dynamics seems odd given what a small proportion of the pool it represents – and it's reduced importance in biological activity.

Response: *We recognize the insignificant role that nitrite plays in biological activity, but what is interesting here, which we have also captured, is its contribution to nitrate over time.*

Figure 2 legend, line 308: delete 'on'

Response: Fixed

Journal Name: Proceedings of the Royal Society B

Appendix C

RSPB-2019-3000.R2: Wild mammalian herbivores are distinct from domestic livestock in their resource subsidies and ecosystem effects

Comments and guidelines by Editor, Proceedings B, Professor Loeske Kruuk

Response: We have followed the guidelines provided in preparing the revised manuscript and relevant files needed for accepted manuscripts

Comments by Associate Editor: Board Member

Thank you for your hard work on this manuscript and the response to reviewers. The manuscript has greatly improved based on both your work and the careful attention by the reviewers. I was a bit disappointed by the authors deciding that the uncertainty estimation was too difficult, but I recognize there are limitations to the data sources that may preclude that at this stage.

Response: We agree and appreciate the comments of reviewers which significantly improved the quality of the ms.

The one point I noticed on re-reading is that towards the end of the discussion, the conceptual picture of this area of the literature does not come precisely into focus, as well as it does in the introduction. For example the line 416: "More studies are needed to assess the spatial and temporal dynamics of dung input into aquatic ecosystems by not only cattle, but other livestock such as goats and sheep, and assess in situ ecosystem responses to these inputs." Yes, of course more empirical studies are always needed, but it would be good at this stage of the manuscript to more precisely define the goal of this area of the literature. What synthetic understanding do we hope emerges from this (and other similar studies) in 5 years time? The anthropogenic change to the world's herbivores has been huge both in the amount and the species composition. Two recent papers may be relevant for this big picture framing:

Bar-On, Yinon M., Rob Phillips, and Ron Milo. "The biomass distribution on Earth." *Proceedings of the National Academy of Sciences* 115.25 (2018): 6506-6511.

Berendes, David M., et al. "Estimation of global recoverable human and animal faecal biomass." *Nature Sustainability* 1.11 (2018): 679-685.

Response: We appreciate the comment and suggestion by the editor. We have strengthened the conclusion by highlighting what we consider to be an emergent topic that also benefits from this study – that of species introductions and/ or rewilding. There has been a suggestion to species introductions or rewilding defaunated habitats by modern analogues, but this study suggests that these introductions may have unintended consequences. We have indicated the need for continued research on ecological effects of introductions of large mammalian herbivores and displacement or replacement of native populations by anthropogenic change. We have edited and added the following text to reflect the change: "Taken collectively, our results expand the current understanding of the role played by large herbivores in the functioning of aquatic ecosystems in African savannas. They also emphasize the species-specific nature of many of these ecological roles, and suggest that species introductions and rewilding efforts seeking to replace extinct species may have unintended outcomes [51, 52]. Therefore, there is a need for more research on the ecological consequences of introduced large herbivores and replacement of native populations by anthropogenic change.

NB: Because the length of the revised manuscript (RSPB-2019-3000.R2), exceeded ten (10) pages, this revised draft has been shortened by 700 words, from 7901 words to 7200 words.